# Projected climate-driven changes in pollen emission season length and magnitude over the continental United States

Yingxiao Zhang [1✉] & Allison L. Steiner [1✉]

Atmospheric conditions affect the release of anemophilous pollen, and the timing and magnitude will be altered by climate change. As simulated with a pollen emission model and future climate data, warmer end-of-century temperatures (4–6 K) shift the start of spring emissions 10–40 days earlier and summer/fall weeds and grasses 5–15 days later and lengthen the season duration. Phenological shifts depend on the temperature response of individual taxa, with convergence in some regions and divergence in others. Temperature and precipitation alter daily pollen emission maxima by −35 to 40% and increase the annual total pollen emission by 16–40% due to changes in phenology and temperature-driven pollen production. Increasing atmospheric $CO_2$ may increase pollen production, and doubling production in conjunction with climate increases end-of-century emissions up to 200%. Land cover change modifies the distribution of pollen emitters, yet the effects are relatively small (<10%) compared to climate or $CO_2$. These simulations indicate that increasing pollen and longer seasons will increase the likelihood of seasonal allergies.

[1] Department of Climate and Space Sciences and Engineering, University of Michigan, Ann Arbor, MI, USA. ✉email: yingxz@umich.edu; alsteine@umich.edu

A nemophilous (or wind-driven) pollen plays an important role in plant fertilization and gene dispersal[1], alters climate by interacting with clouds and radiation[2–4], and triggers allergic diseases such as allergic rhinitis (also known as hay fever) and asthma[5,6]. Pollen-induced respiratory allergy affects up to 30% of the world population, particularly children <18 years old[5,7], and is a worldwide health concern resulting in large economic loss because of medical expenditures, missed work and school days, and early deaths[6,8]. Because pollen emission is closely associated with environmental drivers, climate change could influence pollen emission and consequently the incidence of allergic disease[9,10]. Longer and more intense pollen seasons have been observed over the past few decades[11–13], which is expected to contribute to the exacerbation and aggravation of pollen allergic rhinitis and asthma[7,8,14].

Pollen emissions from anemophilous vegetation are directly correlated with meteorological conditions, such as temperature and precipitation[15,16]. Temperature impacts the number of winter chill hours and spring frost-free days and is strongly associated with the timing of pollen seasons, including the start date, peak emission date, end date, and duration[12,17–19]. Over the past several decades, warmer temperatures have been observed to drive earlier (3–22 days) pollen season start dates[11,20,21] for spring-flowering taxa (e.g., deciduous trees such as *Betula*, *Quercus*, and *Acer*), while late-flowering taxa (*Artemisia* and grasses, which dominate in summer and fall) have delayed pollen season start dates by up to 27 days[11,17]. Prolonged pollen seasons have been recorded for both trees and weeds including *Quercus*, Cupressaceae, Oleaceae, Urticaceae, and Asteraceae[11,12,22]. Precipitation exerts both short-term and long-term effects on pollen emissions. Heavy short-term precipitation significantly reduces atmospheric pollen concentrations via wet deposition[15,23,24], while changes in long-term accumulated precipitation may favor or disadvantage plant growth and therefore alter the total pollen production[25]. In the future, temperature and precipitation are projected to change heterogeneously across the United States (US)[26], and both driving climate variables could directly affect future US pollen emission change patterns. Moreover, the distribution and composition of plant communities are likely to change in the future due to the climate change[27], and further influence the corresponding pollen emission.

Increasing atmospheric $CO_2$ concentrations can fertilize vegetation, enhancing photosynthetic capacity and likely increasing pollen production[28,29]. Higher $CO_2$ concentrations have been observed to increase both the quantity of male flowers[30] and their allergenic protein contents[31], therefore leading to higher pollen and pollen allergen production. Under laboratory conditions, doubling $CO_2$ concentrations increased pollen production by a broad range of 60–1299%[31–34], however, these studies are constrained to chamber experiments for limited species. Despite the uncertainty of how $CO_2$ increases will affect plants in the real world, these studies suggest that increasing atmospheric $CO_2$ concentrations may elevate the intensity of pollen production and increase the prevalence and severity of pollen allergic disease and associated health burdens[14].

While prior observations indicate that pollen phenology is responding to climate change[11,35], large uncertainties remain because pollen observations are sparse in both space and time[12]. Previous observation-based studies are typically limited to small spatial scales[10,12,36] (e.g., in a single city or limited sites) or temporal scales[17,37] (e.g., ~10 years). Existing continental-scale (e.g., the United States, Europe) studies of future pollen emissions only include individual taxa or a small subset of allergic pollen taxa[7,8,16,38–40] or limited climatic drivers[7,20,39]. Because the impacts of climatic drivers on pollen emission vary with vegetation type[22,41] and the dominant pollen taxa vary among regions[42], studying the taxa-specific pollen emission changes is necessary to disentangle the complex influences of climatic drivers on pollen emissions. Here we use a pollen emission model that simulates multiple taxa of pollen emissions at the continental scale[41]. Using a suite of future climate data from the Coupled Model Intercomparison Project version 6 (CMIP6)[43] for two different emissions scenarios[44] including the Shared Socioeconomic Pathways (SSP) 245 and 585 (see "Methods"), we project the change of pollen emissions at the end of the century (2081–2100) compared to the historical period (1995–2014) over the United States for 13 of the most prevalent airborne pollen taxa. As increased $CO_2$ concentration and land cover changes could be important drivers of pollen production in the future[38,40], we also test how rising $CO_2$ concentrations and species range shifts may influence emissions of pollen.

## Results

**Phenological shifts driven by future warmer temperatures.** Using the Pollen Emissions model for Climate Models (PECM)[41], we simulate the daily pollen emissions when including the effects of future temperature and precipitation (without effects of $CO_2$; $\gamma_{CO_2} = 1$ in Eq. (1); "Methods"). Pollen emission phenology (defined with the pollen season start day of the year (sDOY) and end day of the year (eDOY)) is estimated directly from temperature (see "Methods"), driving three categories of change in the pollen season at the end-of-century due to greenhouse gas warming (Fig. 1, and conceptualized in Fig. 2a–c). The first category (Category 1) represents a shift of both the future flowering season sDOY and eDOY to an earlier date than the

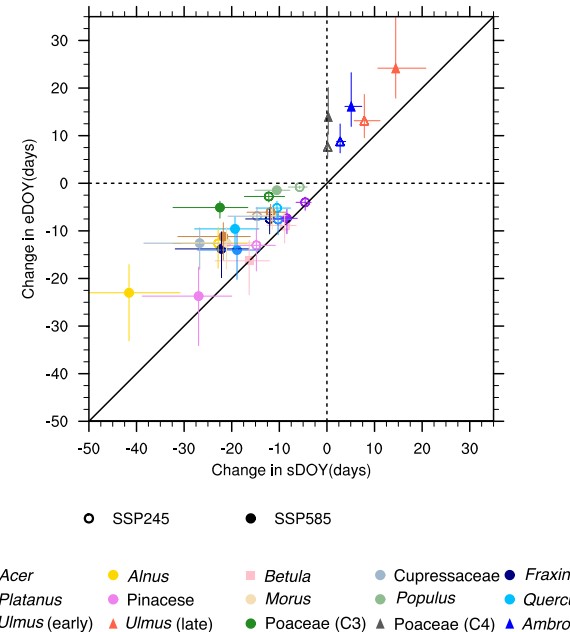

**Fig. 1 Future change in pollen season start date and end date.** The projected change of start day of year (sDOY) and end day of year (eDOY) of pollen season at the end of century (2081–2100) for individual taxa. Each symbol represents the multi-model mean of the spatially and temporally averaged 15 model members. Error bars represent the minimum and maximum from the 15 CMIP6 (Coupled Model Intercomparison Project Phase 6) model ensembles (*n* = 15). Open (Shared Socioeconomic Pathway (SSP) 245) and closed (SSP 585) symbols indicate the results from different future emissions scenarios. Circles represent the taxa in Category 1, squares in Category 2, and triangles in Category 3 (visualized in Fig. 1). 1:1 reference in provided with the solid black line. Source data are provided as source data file.

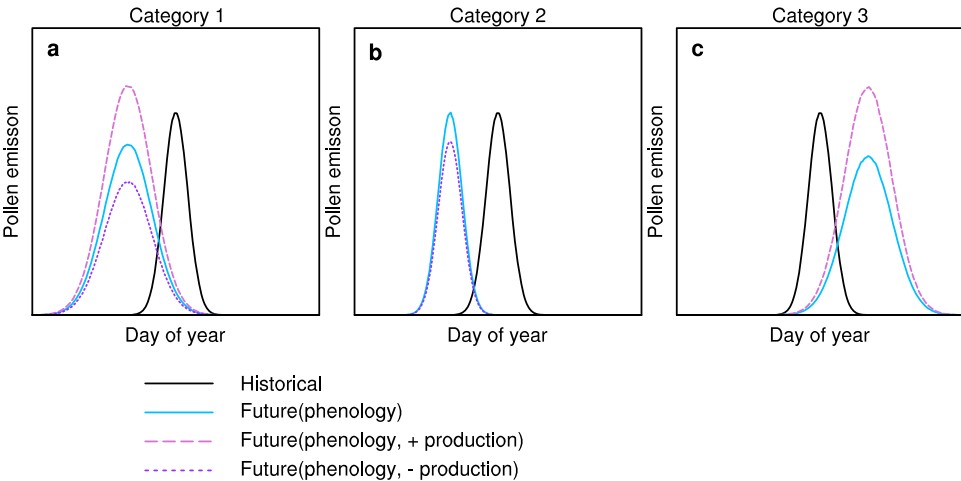

**Fig. 2 Three categories of pollen emission changes. a–c** Conceptual schematic for three categories of pollen emission changes, comparing the historical (black) and future (colors) daily pollen emission. Blues lines indicate the future pollen emission change with only temperature-dependent phenology. Pink and purple curves represent future pollen emission change with phenology change and temperature-dependent changes in the annual production factor ($pf_{annual}$), which can increase (pink) or decrease (purple) production.

historical period (circles in Fig. 1), where sDOY has a stronger temperature dependence than eDOY thereby increasing pollen emission duration (Fig. 2a). Most of the deciduous tree genera (*Acer, Alnus, Fraxinus, Morus, Platanus, Populus, Quercus*, spring-flowering *Ulmus*), both coniferous tree families (Cupressaceae, Pinaceae), and $C_3$ grasses (Poaceae) exhibit Category 1 phenological changes. Absent changes in the pollen production ($pf_{annual}$ in Eq. (1)), this asymmetric increase in the sDOY and longer duration would flatten the pollen emission curve and decrease maximum daily pollen emissions ($E_{pol,max}$) (Fig. 2a). Because temperature is the driving factor of pollen phenology, the spatial distribution of historical duration for individual taxa in Category 1 (Supplementary Fig. 1b) and their future changes (Supplementary Fig. 1f) correspond to the historical spatial temperature pattern (Supplementary Fig. 1a) and projected temperature changes from the CMIP6 models (Supplementary Fig. 1e).

The second category (square in Fig. 1) only includes one genus (*Betula*), where both sDOY and eDOY have a similar temperature dependence and shift earlier at approximately the same rate (Fig. 2b). Therefore, the *Betula* pollen season duration stays approximately the same with no impacts on the $E_{pol,max}$ (Fig. 2b), and the duration change is decoupled from temperature (e.g., *Betula* in Supplementary Fig. 1g).

Vegetation in the third category includes short-day species[45,46], with maximum emissions in the late summer and early fall (e.g., late-flowering *Ulmus*, $C_4$ grasses (Poaceae), and *Ambrosia*) that flower as days shorten in the Northern Hemisphere. Despite seasonal light as a driver, sDOY and eDOY for these taxa was best predicted with temperature[41]. Both sDOY and eDOY increasing with warmer temperatures (triangles in Fig. 1) and are projected to occur later under end-of-century conditions, with a stronger eDOY temperature dependence than sDOY resulting in a longer duration (Fig. 2c). Taxa in this category experience a longer duration that flattens the pollen emission curve when not considering pollen production changes (Fig. 2c). Similar to Category 1, the duration changes in Category 3 are closely associated with temperature change, with greater increases in duration in the north (up to 16 days) than in the south (up to 12 days) (e.g., *Ambrosia* in Supplementary Fig. 1h).

Temperature effects on simulated pollen phenology vary substantially for different taxa, leading to a broad range in the increase of pollen season duration. Under scenario SSP 585 (see

"Methods"), the duration increases in Category 1 are 2–19 days, where the large variability is a function of the 11 different vegetation taxa in this category (Fig. 1; circles). For the short-day vegetation in Category 3, the duration increases from 10 to 14 days (Fig. 1; triangles). The only genus (*Betula*) in Category 2 does not exhibit any duration changes as its sDOY and eDOY shift by the same amount (Fig. 1, squares). In addition, the end-of-century sDOY and eDOY changes are a function of future emissions scenarios. Because temperature is the sole driver of modeled pollen phenology change, duration shifts are consistent with the projected temperatures. The duration change with high-end scenario SSP 585 (2–19 days) is approximately twice of that with moderate emission scenario SSP 245 (1–10 days), corresponding to the temperature changes of 4–6 K degrees and 2–3 K, respectively (Supplementary Fig. 2c, e).

**Changes in maximum daily pollen emission driven by future climate.** In addition to altering phenology, warmer temperatures will also impact vegetation physiology, either facilitating or restraining growth and impacting plant biomass. These changes have the potential to increase or decrease the annual total pollen production[37,47] ($pf_{annual}$; Eq. (5)). Limited observational data suggest that many taxa are projected to have higher pollen production with warmer temperatures (e.g., *Alnus, Morus, Platanus, Quercus*, spring and late-flowering *Ulmus*, Cupresseceae, *Ambrosia*, C3 and C4 Poaceae3), while some taxa will have lower pollen production (e.g., *Acer, Betula, Fraxinus, Populus*, Pinaceae) (Supplementary Table 1 and Supplementary Fig. 3)[37,48]. The variations in $pf_{annual}$ in conjunction with phenology can influence the maximum daily pollen emission ($E_{pol,max}$; grains $m^{-2} d^{-1}$), which is a potential health exposure metric. Increasing $pf_{annual}$ under future climate typically increases the $E_{pol,max}$ and counteracts the lengthening season duration for Category 1 and 3 vegetation (Fig. 2a, c, respectively), especially for taxa with a strong dependence on temperature (e.g., *Alnus, Platanus, Quercus*, late-flowering *Ulmus*, Cupresseceae, *Ambrosia*; Supplementary Fig. 3). In contrast, decreasing $pf_{annual}$ will further decrease the $E_{pol,max}$ (Fig. 2a, b), with greater decreases for taxa projected to have longer durations in the future (*Acer, Fraxinus, Populus*, Pinaceae; Fig. 2a).

As both pollen production and phenology have the potential to impact pollen emission maxima ($E_{pol,max}$), a sensitivity analysis (see "Methods") indicates that the temperature-dependent

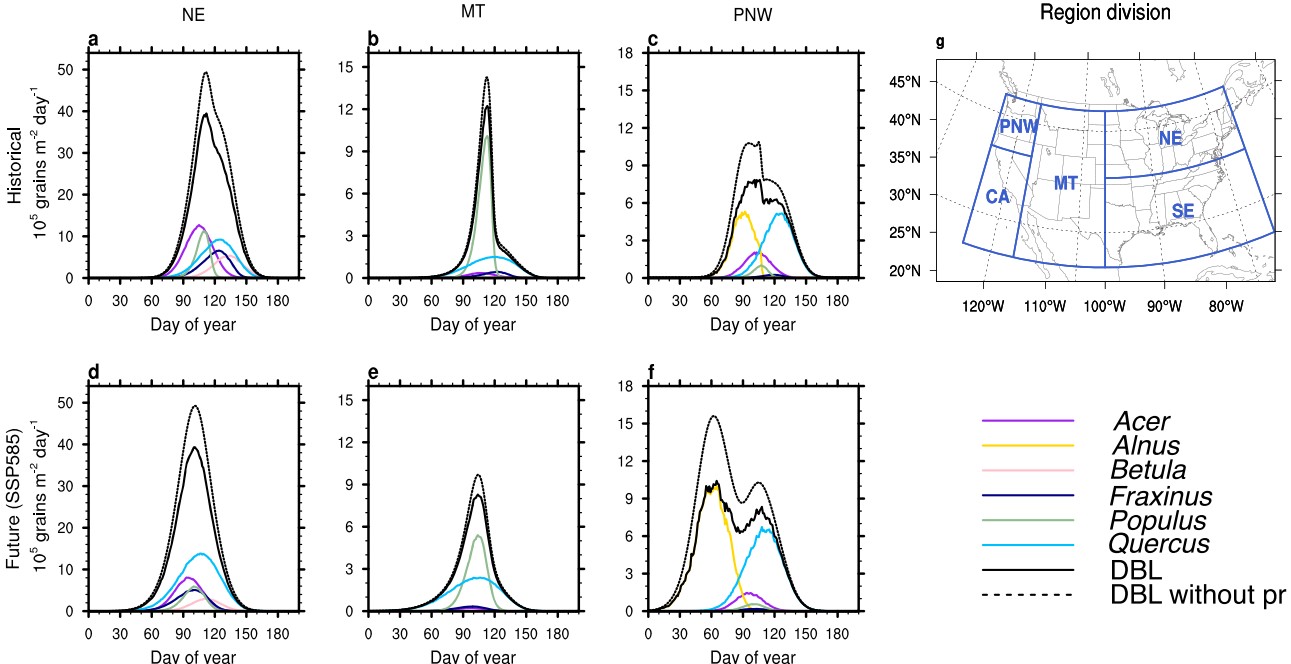

**Fig. 3 Simulated regional pollen seasonal magnitude and timing in the historical and future. a–f** Twenty-year average time series of daily pollen emission flux (grains m$^{-2}$ d$^{-1}$) of the six dominant individual tree taxa in the deciduous broadleaf forest (DBL) and the total DBL emission with (solid black line) and without (dashed black line) precipitation effects. **a–c** Historical (1995–2014) emission and (**d–f**) end of the century (2081–2100) emissions from multi-model average simulation for shared socioeconomic pathway (SSP) 585. **g** We define five geographic regions: Northeast (NE; 38–48° N and 70–100° W), Southeast (SE; 25–38° N and 70–100° W), Mountain (MT; 25–48° N and 100–116° W), California (CA; 25–40° N and 116–125° W) and Pacific Northwest (PNW; 40–48° N and 116–125° W). Three regions are selected for DBL daily pollen emission analysis: Northeast, NE (**a**, **d**); Mountain, MT (**b**, **e**); Pacific Northwest, PNW (**c**, **f**).

regression and normalization parameters of pollen production ($m_{prod}$, $b_{prod}$, and $P_{norm}$ in Eq. (5)) have the greatest impacts on the simulated pollen amount (Supplementary Table 2). For taxa with pollen season duration sensitive to temperature change (*Alnus, Platanus, Populus*, late-flowering *Ulmus*, C4 grass), the regression parameters of phenology can also become important. Overall, the magnitude of annual pollen production is one of the most important parameters in model simulation ($pf_{annual}$ in Eq. (5) and Supplementary Table 2).

Phenological shifts can create new overlap between different taxa in the future and thereby alter the total $E_{pol,max}$, with regional variations dependent on the vegetation taxa. In the Northeast (NE; Fig. 3g), many deciduous broadleaf (DBL) taxa (e.g., *Acer, Betula, Fraxinus*, and *Populus*) project lower $E_{pol,max}$ driven by temperature-based $pf_{annual}$ reductions and longer pollen season durations (except *Betula* that is driven only by $pf_{annual}$). For the three dominant NE DBL genera (*Quercus, Populus*, and *Acer*) in the historical period, *Acer* pollinates first around DOY (day of year) 70, with a peak magnitude of $12 \times 10^5$ grains m$^{-2}$ on DOY 100. This is followed by *Populus* with a sDOY of 80 and peak magnitude of $11 \times 10^5$ grains m$^{-2}$ on DOY 110, and *Quercus* with a sDOY of 80 and peak magnitude on DOY 130 of $10 \times 10^5$ grains m$^{-2}$ (Fig. 3a). In the future, the $E_{pol,max}$ increases $4 \times 10^5$ grains m$^{-2}$ (40%) for *Quercus* but decreases $4 \times 10^5$ grains m$^{-2}$ (33%) and $5.5 \times 10^5$ grains m$^{-2}$ (50%) for *Acer* and *Populus*, respectively (Fig. 3d). These three taxa shift to earlier sDOY with longer season durations, driving a convergence of the flowering season. However, the temperature-driven changes in production lead to minimal change in future total $E_{pol,max}$ of DBL (about $5 \times 10^6$ grains m$^{-2}$ on DOY 105 for the historical and future). Similarly in the Mountain (MT) region, the two main taxa (*Quercus* and *Populus*) both shift to earlier phenologies and their maximum pollen emissions converge, but the total pollen is dominated by

the reduction of *Populus* $pf_{annual}$, driving a decrease of total $E_{pol,max}$ of DBL (Fig. 3e). However, the convergence of phenology for the evergreen needleleaf (ENL) vegetation families (Pinaceae and Cupressaceae) leads to an increase of ENL total $E_{pol,max}$ (Supplementary Fig. 4).

In contrast, the phenology of the two dominant emitters in Pacific Northwest (PNW) (*Alnus* and *Quercus*) diverge. *Alnus* emissions start on DOY 60 and overlap with *Quercus* around DOY 100 in the historical period (Fig. 3c). In the future, *Alnus* sDOY shifts earlier at a faster rate than *Quercus*, thereby reducing coincident emissions (Fig. 3f). Both *Alnus* and *Quercus* $pf_{annual}$ are positively impacted by warmer future temperatures, yet the divergence of phenology mitigates the production increases and results in a modest 30% increase of the total $E_{pol,max}$ in the PNW.

Future precipitation changes can also influence pollen emissions. In the historical period, precipitation decreases the mean daily pollen emissions by up to 30% (Fig. 3a–c), especially in the regions with higher daily precipitation intensity (NE, SE, and PNW, Supplementary Fig. 2b). In the future, monthly averaged daily precipitation is projected to increase up to 30% during spring and winter (Supplementary Fig. 5), with the most significant changes occurring in NE and PNW under SSP 585 (Supplementary Fig. 2f). Because spring is the flowering season for most high-emitting anemophilous vegetation, increased future precipitation decreases $E_{pol,max}$ up to 40% of in the NE and PNW regions (Fig. 3d, f).

Overall, temperature and precipitation future climate effects alter $E_{pol,max}$ from −35 to 40% under the future SSP 585 scenario (Fig. 4j), driven by the competing effects of duration lengthening, $pf_{annual}$ change, and precipitation scavenging. The $E_{pol,max}$ of DBL increase in the southern regions by 10–30% but are reduced up to 20% in the NE and 40% MT regions (Fig. 4f). Due to the greatest magnitude of warming in the north (Supplementary Fig. 2e),

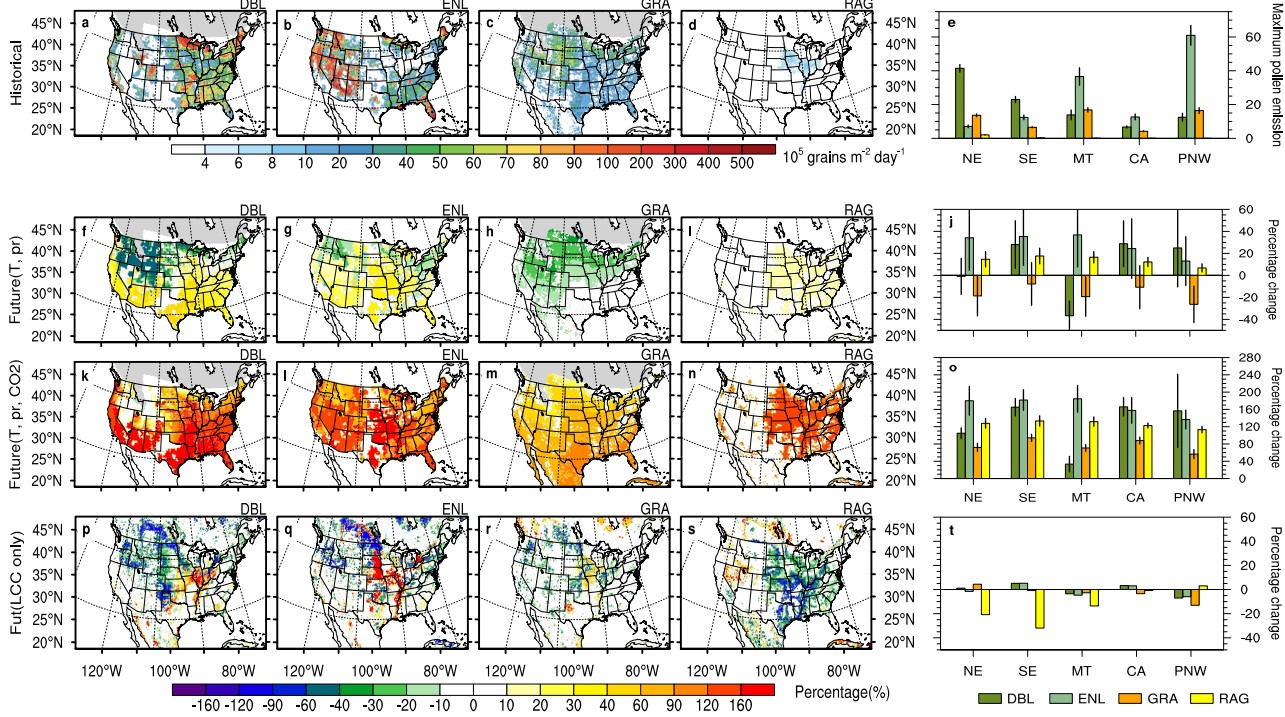

**Fig. 4 Historical and future changes of maximum daily pollen emissions ($E_{pol,max}$). a–e** Multi-model historical average (1995–2014) maximum daily pollen emission flux ($E_{pol,max}$) over the United States (grains $m^{-2}$ $d^{-1}$). **f–j** Projected multi-model average future $E_{pol,max}$ change (%) at the end of century (2081–2100) for shared socioeconomic pathway (SSP) 585, with the effects of temperature (T) and precipitation (pr) only, and (**k–o**) projected future $E_{pol,max}$ change (%) due to temperature, precipitation, and $CO_2$. Panels **a–o** use the taxa-based pollen emission model (PECM) driven by meteorology input data from each CMIP6 model to calculate the multi-model average. **p–t** Plant functional type (PFT)-based model $E_{pol,max}$ change (%) with land cover change (LLC) effects only. The simulation is conducted using PFT-based pollen emission model (PECM) with historical (2015) and future (2100) PFT land cover and driven by the multi-model average climate input data. Columns represent different PFTs: deciduous broadleaf forest (DBL) (**a, f, k, p**), evergreen needleleaf forest (ENL) (**b, g, l, q**), grasses (GRA) (**c, h, m, r**), ragweed (RAG) (**d, i, n, s**). Bar charts (**e, j, o, t**) show the spatial averages in five subregions (Fig. 3g) with error bars representing the standard deviation from the average of 15 independent CMIP6 (Coupled Model Intercomparison Project Phase 6) model ensembles ($n = 15$) in each region (**e, j, o**).

$E_{pol,max}$ decreases are caused by both duration lengthening and changes in the dominant taxa $pf_{annual}$, as several northern DBL taxa exhibit a negative relationship with temperature (Supplementary Fig. 3a, c, d, g). ENL $E_{pol,max}$ increase 10–40% over most of the US with some decreases in high latitude and mountain regions (Fig. 4g). The geographic distribution and composition of plant community plays an important role of the spatial differences of DBL and ENL pollen emissions, with the interactions between different taxa phenology and effects of temperature on $pf_{annual}$ (Supplementary Fig. 3). Emissions from grasses (Poaceae) and ragweed (Ambrosia) lack the intra-PFT interaction in pollen phenology. Therefore, the future $E_{pol,max}$ change for grasses and ragweed are determined by the competing effects of the temperature-based $pf_{annual}$ and pollen season duration changes. For grasses, the longer duration dominates and decreases $E_{pol,max}$ by 10–40% over all regions, with greater reductions in the north due to the larger temperature increases and longer season durations (Fig. 4h). In contrast, ragweed exhibits a stronger dependence of $pf_{annual}$ with temperature (Supplementary Fig. 3m) and future $E_{pol,max}$ increases about 10–20% over the continental US (Fig. 4i).

**Climate-driven increases in annual total pollen emissions.** In addition to the maximum daily pollen emission ($E_{pol,max}$), future climate also impacts the simulated annual total pollen emission ($E_{pol,ann}$; the sum of simulated daily pollen emission). The

magnitude of $E_{pol,ann}$ is impacted by $pf_{annual}$, precipitation, and land cover fraction of each vegetation types. Due to the variation in vegetation geographic distribution, the composition and contribution of $E_{pol,ann}$ from different taxa vary significantly across the US (Fig. 5). In the historical, $E_{pol,ann}$ is the highest in the PNW ($4.2 \times 10^8$ grains $m^{-2}$ $yr^{-1}$), because of the extensive vegetation coverage as well as large pollen production of the dominant taxa (Pinaceae, Cupressaceae). NE has relatively lower $E_{pol,ann}$ ($2.7 \times 10^8$ grains $m^{-2}$ $yr^{-1}$), with a greater number of contributing taxa that is dominated by *Acer, Populus,* and *Quercus*.

Assuming the land cover fraction and $CO_2$ concentration are the same as the historical period, future pollen emission composition from different taxa changes substantially among the regions (e.g., relative contributions from *Quercus* increase up to 8% in the NE, and Cupressaceae increase up to 10% in the MT and PNW in the future; Fig. 5). While temperatures are warming, the total $E_{pol,ann}$ over NE and PNW regions show relatively small increases (16 and 26%, respectively) as the $pf_{annual}$ of the several regional taxa exhibit negative correlations with temperature (e.g., *Acer, Betula, Populus* and Pinaceae; Supplementary Fig. 3a, c, g, l). In contrast, the SE and MT regions have larger increases in $E_{pol,ann}$ (up to 40%), where the dominant taxa (*Quercus* and Cupressaceae) have a strong positive dependence on temperature (Supplementary Fig. 3h, k).

Overall, simulated $E_{pol,ann}$ increases 16–40% over the United States when considering future climate effects only (Fig. 5). Because $E_{pol,ann}$ integrates pollen emissions over the entire year,

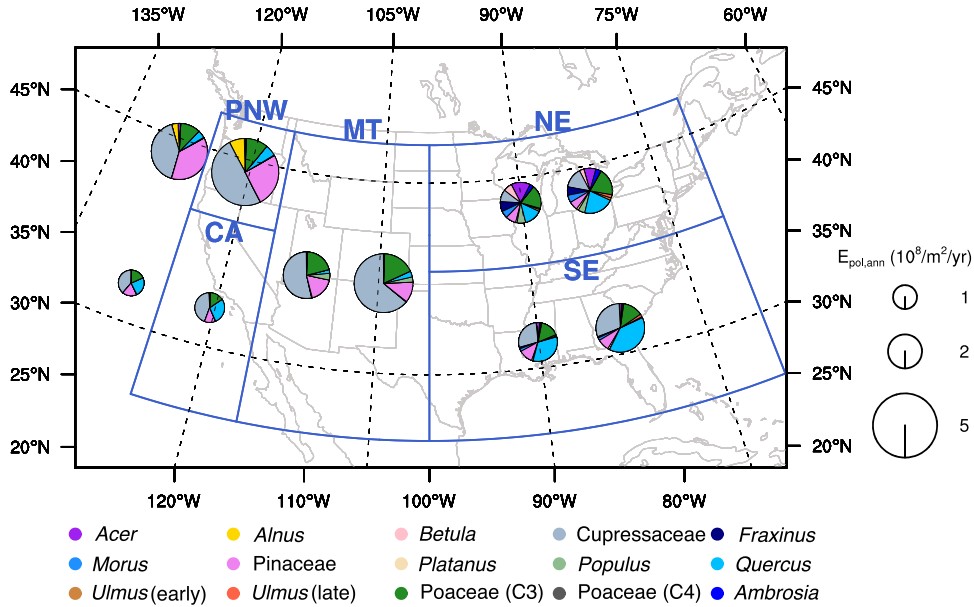

**Fig. 5 Average annual total pollen emission in the historical and future.** Regional average annual total pollen emission ($E_{pol,ann}$, unit: $10^8$ grains $m^{-2} yr^{-1}$) in the historical (left; 1995-2014) and future shared socioeconomic pathway (SSP) 585 scenario (right; 2081-2100) for the 5 geographic regions (Northeast (NE), Southeast (SE), Mountain (MT), California (CA), and Pacific Northwest (PNW). Chart size indicates the amount of total pollen emitted and different colors represent the contribution of different taxa. The data used here are provided as source data file.

its magnitude is driven by the $pf_{annual}$ and precipitation, and phenological shifts do not impact the integrated total. Therefore, although the pollen emission peak ($E_{pol,max}$) decreases over some high latitude and Mountain (MT) regions for DBL, ENL, and grasses (Fig. 4f–h) due to phenological shifts, the temperature-dependent $pf_{annual}$ and precipitation are the drivers for an increase in total accumulation ($E_{pol,ann}$) (Fig. 5).

**$CO_2$ and land cover effects on pollen emissions.** When we allow future $CO_2$ to affect pollen production ($\gamma_{CO_2} = 2$ in Eq. (1); "Methods"), $E_{pol,ann}$ increases up to 250% and $E_{pol,max}$ increases by up to 200% throughout the continental U.S. (Fig. 4k–o). This suggests that increased $CO_2$ could counteract the simulated daily maximum pollen emission decreases due to climate-only effects in some regions. For DBL and grasses, the increases are spatially consistent with temperature changes and are greatest in the South (Fig. 4k, m). ENL and RAG increase throughout the US (Fig. 4l, n), with ENL affected by the convergence of intra-taxa phenologies (Supplementary Fig. 4). Because multiple taxa and subsequent overlap changes are not simulated for GRA and RAG, their regional variations are related to the spatial distribution of future temperature and precipitation changes (Supplementary Fig. 2). With warmer projected temperatures in the south and decreased precipitation intensity in the Southwestern US, the future maximum emissions for GRA are about 30% greater in the south than in the north (Fig. 4m, o). For RAG, the pollen maximum emission in the southern latitudes is also slightly higher (10%) than northern latitudes (Fig. 4n, o).

Future climate change and anthropogenic impacts are likely to shift the spatial distribution of plant communities[27] and therefore impact pollen emissions. Because gridded taxa-specific land cover change data are not available, we test the impact of land cover change using projections of plant functional types (PFTs) from GCAM-Demeter land use dataset[49] (Supplementary Fig. 6). PECM1.0 can estimate pollen emissions based on PFT with greater uncertainties than the taxa-specific method employed for the previous simulations[41]. We simulate the future maximum pollen emission with PFT-based pollen emission model using

both the historical and future land cover (see "Methods") (Supplementary Fig. 7). The difference between the two PFT simulations indicates the impact of future land cover change on maximum pollen emissions over the US, noting that the $pf_{annual}$ vary between the PFT and taxa-based models (Supplementary Table 1).

Future PFT changes projected by the Global Change Analysis Model (GCAM)[49] simulate an increase in tree coverage in the Central US and Mississippi River Valley at the expense of crop, and some decrease in tree coverage at high altitudes of the Rockies and the Pacific Northwest (Supplementary Fig. 6). Compared to tree PFTs, changes to grassland are relatively small and occur in smaller patches (Supplementary Fig. 6c). Ragweed coverage is based on urban and crop land cover, and the future reduction of the cropland due to the expansion of grasses or trees[49] drives large decreases (up to 80%) of potential ragweed land cover over the eastern US (Supplementary Fig. 6d). For the two dominant US tree types (DBL and ENL), the regional pollen maximum emission increases up to 6% in SE and CA (California) while decreasing up to 7% in the MT and PNW regions (Fig. 4p, q, t). Future changes to grass cover change grass pollen emissions by -18–5% (Fig. 4t). Ragweed emissions have the largest pollen emission maxima decreases over NE and SE (up to 32%; Fig. 4t) because of cropland reduction, however, we note large uncertainties in the spatial distribution of ragweed. Compared to emission changes due to climate or $CO_2$ effects, the maximum pollen emission changes due to land cover changes at the regional scale are relatively small (−32 to 6%).

## Discussion
Taken together, we project that under the SSP 585 future climate scenario the pollen season will start earlier (up to 40 d) and become longer (+19 d) with temperature change, and the annual total pollen emission will also increase (16–40%) over the United States. Climate-only driven changes are relatively small (−35 to 40%) compared to large maximum emission increases (up to 200%) when accounting for increased $CO_2$ on pollen production (Fig. 4j, o), although we note the large uncertainties in the $CO_2$

effects on production[50]. Land cover change can either increase or decrease the future pollen emission maxima, but the regional impacts are smaller (−32 to 6%) than other factors and we conclude that the land cover change influence on pollen emission is likely to have less of an impact than the influence of meteorological factors or $CO_2$.

These projected trends correspond to previous observational studies based on ~30-year historical data analysis, which have identified a 20 d advance, 8 d lengthen of pollen season[13], a 46% increase of annual total pollen emission, and a 42.4% enhance of peak pollen emission[11]. While these prior studies have evaluated total pollen changes in the historical period, this work highlights the importance of studying taxa-specific pollen emission and finds that the influence of climate change on daily pollen emissions varies for different regional forest compositions. We also demonstrate that climate could drive the convergence or divergence of individual taxa pollination, which can magnify or mitigate the climate change impacts and have significant implications for evaluating the consequences of future pollen emissions. While pollen is one of the main causes of human allergies, these projected future changes may lead to growing population's exposure and the severity of symptoms in individuals with allergic rhinitis and allergic asthmas induced by pollen.

One important limitation to this study is the uncertainties associated with the pollen emission parameterizations. PECM is based on geographically constrained historical pollen counts to derive the relationships between pollen and climate, and sparse data coverage with an urban site focus may limit our ability to parameterize pollen emissions from natural forests[41]. The sensitivity analysis of model parameters (see "Methods") indicates the dominant uncertainty is related to the pollen production and the climate-relevant production parameters, which is derived from a limited suite of field-based studies (Supplementary Table 1). More measurements across space and time could improve our understanding of pollen production and better constrain the model simulations. In addition, limited experimental chamber data for a small number of vegetation taxa[31–34,38] create uncertainty in the parameterization of the effects of higher $CO_2$ concentrations on pollen production. We broadly applied a doubling of pollen production for the end-of-century $CO_2$ concentrations projected by the SSP 585 scenario, and additional studies that examine the role of $CO_2$ and climate on pollen emission interannual variability are greatly needed. Finally, large uncertainties in future plant community shifts[51,52] also limit the simulations of land cover change effects on pollen emission. Recent advances in species distribution modeling include the development of new approaches (e.g., regression-based and machine learning)[53], yet there are large uncertainties connected to climate change projection and biotic stresses (e.g., insects, fungi, bacterial)[53]. Gridded taxa-specific land cover change data for multiple taxa over the entire CONUS is still lacking[54–56]. Our simulations using PFT-based land cover change data provide overall estimates of vegetation shifts, but the development of the spatially resolved taxa-specific land cover data over a large scale will be crucial to evaluate the effects of plant community composition change on future pollen emission.

Despite these limitations, this study quantifies the potential climate change impacts from $CO_2$, temperature, and precipitation on pollen emission over the US. In the historical period, temperature is the dominant driver of continental-scale pollen emissions and the $CO_2$ effect is relatively small with about 50 ppm increase in the past 30 years[13]. However, in the future, $CO_2$ concentrations are projected to increase dramatically, especially under the high-emission scenario utilized here (e.g., an increase of about 700 ppm at the end of the century under SSP 585[57]), and its effects on pollen production have the potential to lead to large

pollen increases. Our approach to simulate individual taxa pollen emissions highlights that changing climate will alter the pollen phenology and total production of individual taxa, potentially increasing the seasonal overlap and overall increasing pollen emissions. Land cover change has relatively smaller effects in our simulations, suggesting that the climate drivers may be more important and occurring faster than the shifts in vegetation distribution, however our approach does not account for the spatial shifts of individual taxa ranges, which will certainly be important to assess future regional pollen emission composition. This study provides an important predictive tool to start to investigate the consequences of climate change on future plant communities and their corresponding health effects.

## Methods

**Pollen emissions model.** We utilize the Pollen Emissions model for Climate Models (PECM1.0[41]), which is a prognostic model developed from historical pollen count data from the National Allergy Bureau (NAB) of the American Academy of Allergy, Asthma and Immunology (AAAAI). It simulates the phenology of the 13 most prevalent wind-pollinating taxa (including *Acer, Alnus, Ambrosia, Betula, Cupressaceae, Fraxinus, Poaceae, Morus, Pinaceae, Platanus, Populus, Quercus,* and *Ulmus*) over the United States, which accounts for 77% of the total pollen counts across the United States during 2003–2010[41]. It predicts pollen emission for a broad range of taxa at a large geographic scale (25-km resolution in this study), and it is able to capture up to 57% of the variance of pollen season.

The pollen emission model simulates the pollen emission flux of individual taxon ($E_{pol}$; grains $m^{-2} d^{-1}$) over the continental United States as a function of land cover and meteorological factors:

$$E_{pol} = A \times pf_{annual} \times \gamma_{phen} \times \gamma_{precip} \times \gamma_{CO_2} \qquad (1)$$

where $A$ is the vegetation land cover fraction ($m^2$ vegetated $m^{-2}$ total area) based on the observed land cover data (see "Model input data" section), and $pf_{annual}$ (grains $m^{-2} d^{-1}$) is the annual pollen productivity factor. Pollen phenology is dictated by $\gamma_{phen}$, which determines the seasonal pollen emission and is empirically calculated with a Gaussian distribution (Eq. (2)):

$$\gamma_{phen} = e^{-\frac{(t-\mu)^2}{2\sigma^2}} \qquad (2)$$

where $\mu$ and $\sigma$ are the mean and half-width of the Gaussian, respectively. They are determined by the pollen season to start day of the year (sDOY) and end day of the year (eDOY):

$$\mu = \frac{sDOY + eDOY}{2} \qquad (3)$$

$$\sigma = \frac{eDOY - sDOY}{a} \qquad (4)$$

The timing of pollen emission (sDOY and eDOY) for individual taxon is linearly related to the previous-year annual average temperature (PYAAT) with a linear regression between observed first or last day of pollen count and the corresponding temperature (e.g., Supplementary Fig. 8), with a constant factor controlling the width (a =3) based on evaluation versus observed pollen counts.

Here we make three modifications to PECM1.0, including (1) a modification to the annual pollen production ($pf_{annual}$), (2) the introduction of a precipitation factor ($\gamma_{precip}$) to account for the reduction in emissions during wet conditions, and (3) a carbon dioxide factor ($\gamma_{CO_2}$) that scales the pollen production to $CO_2$ concentrations.

In PECM1.0, taxa-dependent pollen production values by taxa ($P_{annual}$; grains $tree^{-1}$ converted to grains $m^{-2}$) are derived from limited literature field surveys[47,58–66] and are an important input (see "Model sensitivity analysis"). $P_{annual}$ values are updated from Wozniak et al.[41] to include new literature (Supplementary Table 1). However, in PECM1.0, $P_{annual}$ was held constant for each taxon and does not account for the potential interannual variation of pollen production with time. A few studies have examined the interannual variability of pollen count[67,68], but a complication of using atmospheric pollen count to determine the interannual variability of production is that many meteorological factors influence the count (e.g., wind, precipitation, and other meteorological conditions such as boundary layer height). Therefore we scale the literature production factors ($P_{annual}$) to a temperature-dependent annual production factor ($pf_{annual}$) with the observed linear relationship between log-transformed observed annual total pollen counts and previous-year annual average temperature (PYAAT) for each taxa (Eq. (5)) normalized to the temperature-dependent production ($P_{norm}$):

$$pf_{annual} = \frac{\exp(m_{prod} \times PYAAT + b_{prod})}{P_{norm}} \times P_{annual} \qquad (5)$$

$m_{prod}$ and $b_{prod}$ are the slope and intercept of the linear regression, respectively, of the observed pollen count and temperature and vary for each taxon

(Supplementary Table 1). $P_{norm}$ is the pollen count at the historical average temperature over the US. One exception is for the genus *Alnus*, which is has a limited spatial range in the Pacific Northwest (PNW) and we use the historical average temperature in the PNW.

Nine of the 15 total PYAAT-pollen count relationships are statistically significant ($P < 0.05$, Supplementary Fig. 3), suggesting that this regression captures the influence of prior year temperature on the subsequent year's pollen production. While there are likely many other factors that influence pollen production (e.g., moisture resources, soil nitrogen, etc.), this provides a simple method to account for year over year pollen production. The annual total pollen counts of a majority of taxa (10 of 15) are positively correlated to temperature ($m_{prod} > 0$) (including *Alnus, Morus, Platanus, Quercus*, spring, and late-flowering *Ulmus*, Cupressecease, *Ambrosia*, C3 and C4 Poaceae), suggesting annual pollen production will increase with the warmer temperatures and therefore lead to higher pollen counts. However, several taxa with broad spatial coverage over the US show negative correlations (*Acer, Betula, Fraxinus, Populus*, Pinaceae), with only *Acer* and *Betula* exhibiting statically significant correlations as well as high sensitivity to temperature change ($|m_{prod}| > 0.05$). Overall, the observed relationship between pollen counts and temperature suggests mostly positive correlations with similar statistics demonstrated in Schramm et al.[48], with exceptions for some vegetation that exhibits broad spatial coverage (e.g., *Acer, Populus*, Pinaceae).

Precipitation exerts a dual effect on pollen emission[11]. High precipitation will remove emitted pollen grains from the atmosphere and can prevent pollen release, while low annual precipitation may limit plant growth and may restrain pollen production. However, due to the complexity of precipitation effects, we did not find robust correlations between precipitation and pollen production at longer time scales. Therefore, we only include the precipitation scavenging effects in this study through a precipitation factor ($\gamma_{precip}$). If the precipitation exceeds a minimum intensity of 5 mm d$^{-1}$ [69], all emitted pollen is removed from the atmosphere[15,16,23,24], and the precipitation factor ($\gamma_{precip}$) is set to zero for that day. While other atmospheric conditions such as wind and humidity are linked to precipitation changes and may inversely lead to higher pollen concentration before or during the rainfall events[69,70], this study does not account for meteorological factors other than temperature and precipitation, as this requires coupling with an atmospheric model as in Wozniak et al.[41].

The impact of atmospheric $CO_2$ on pollen emissions is based on laboratory studies for limited taxa, which exhibit a wide range of responses to $CO_2$ increases[31–34,38]. Despite these uncertainties, we consider $CO_2$ effects in the model as an additional sensitivity test to understand the potential of this feedback on future pollen emissions. For the SSP 585 emissions scenario at the end of the century, we conduct an additional simulation where $\gamma_{CO_2} = 2$, based on the previous studies[31–34,38,50], which essentially assumes a doubling of the pollen production factor for all taxa.

**Model input data**. PECM can calculate pollen emissions based on two different types of land cover: taxa-based land cover or plant functional type (PFT)-based land cover[41]. Generally, the taxa-specific model showed better agreement with observed pollen counts than the PFT model[41] and we use taxa-based model for the assessment of future climate changes in this work. The taxa-based land cover data of 11 dominant tree taxa are defined by the Biogenic Emission Landuse Dataset version 3 (BELD)[71], with satellite-derived land cover data from the Community Land Model 4 (CLM4)[72] to provide spatial distributions of Poaceae (C3 and C4 categories) and *Ambrosia* (calculated using the urban and crop categories)[41]. While emissions are simulated at the taxa level, for analysis we group the taxa into 4 PFTs: deciduous broadleaf forest (DBL), evergreen needleleaf forest (ENL), grasses (GRA), and ragweed (RAG). DBL contains 9 tree taxa (*Acer, Alnus, Betula, Fraxinus, Morus, Platanus, Populus, Quercus*, and *Ulmus*), and ENL includes two tree families (Cupressaceae and Pinaceae). GRA (Poaceae) has two types (C3 and C4 grass) with distinctly different flowering times, and RAG has one taxon (*Ambrosia*). The emission of each PFT is the aggregate of the modeling emission of all taxa belonging to that PFT.

In the future, the distribution and composition of plant communities can be altered by climate change (e.g., increase the relative abundances of heat-tolerant species, change the distribution of water-demanding species[30]) or human activities (e.g., impact land cover change and seed dispersal[44]), yet the effects of climatic drivers on plant communities are poorly understood and further constrain predictive model development[55,56]. Due to the large uncertainties and the lack of available data for taxa-level spatial distribution shifts, we include a sensitivity test of future land cover change effects on pollen emission using the PFT-based PECM model[41]. In the climate-based simulations of this study with the taxa-based model, we do not consider the changes to the land cover and land use data in the future and hold vegetation land cover in the model constant. For the sensitivity test, we simulated the future maximum daily pollen emission both using historical (2015) and future (2100) PFT land cover data from GCAM-Demeter land use dataset[49], which is driven by the same climate forcing data used for PECM. Compared to the taxa-based model, the PFT version of the model extends to all of North America and simulates higher ENL pollen emissions over the US (Supplementary Fig. 7).

Daily temperature and precipitation data are from 15 models from the Coupled Model Intercomparison Project Phase 6 (CMIP6) (https://esgf-node.llnl.gov/search/cmip6/), with individual model information included in Supplementary Table 3. Climate data are regridded to a 25 km Lambert Conformal Conic projection[41] using the Earth System Modeling Framework (ESMF) higher-order patch regridding method over the United States to match the spatial resolution of PECM. We calculate the previous-year annual average temperature (PYAAT) for each model grid cell from daily temperature as a dependent variable to model the pollen season phenology. Because pollen grains are formed in the year previous to flowering, their amount is determined by the photosynthates accumulated in the past summer and correlated with previous-year annual average temperature (PYAAT)[73]. Pollen emissions are simulated using the meteorology data input from each CMIP6 model and then an evenly weighted multi-model average from 15 PECM simulations is calculated for analysis.

We analyze pollen emissions at a 25-km resolution over the continental US for two periods: the historical (1995–2014) and end-of-century future (2081–2100) and compare differences in pollen season timing and pollen emission magnitude. For the future, we utilize model simulations for two emissions scenarios combining both shared socioeconomic pathways (SSPs) and representative concentration pathways (RCPs), specifically, SSP 245 is the "middle of the road" development pathway (SSP2) with a 4.5 W/m$^2$ radiative forcing level by 2100 corresponding to RCP4.5 scenario, and SSP 585 is the high fossil-fueled development (SSP5) with a higher (8.5 W/m$^2$) radiative forcing level by 2100 (RCP8.5)[44,57].

**Model sensitivity analysis**. To evaluate the uncertainties of the PECM model, we conducted a sensitivity analysis using the Morris method[74]. The Morris method is a "one-at-a-time" approach, allowing a computationally efficient uncertainty evaluation for a large number of model parameters. Nine parameters for each of the fifteen taxa used in the model are studied in this analysis (Supplementary Table 4). The uncertainty ranges of each parameter are determined by literature values ($P_{annual}$) or computed by the 95% confidence level (the linear regression slope (m) and intercept (b) used to calculate start date ($m_{sDOY}$, $b_{sDOY}$) and end date ($m_{eDOY}$, $b_{eDOY}$) of pollen season, and the pollen production ($m_{prod}$, $b_{prod}$)). Because the normalized pollen production ($P_{norm}$) is calculated using the linear regression with $m_{prod}$ and $b_{prod}$, its uncertainty range is also determined by the range of $m_{prod}$ and $b_{prod}$. For the phenological Gaussian width $a$, the maximum and minimum value is obtained by ± 0.2 of the original value (3).

Using the method of Morris from the Sensitivity Analysis Library (SALib) in Python (https://salib.readthedocs.io/en/latest/), we conducted 1000 (N) model runs for each taxon, where the $N$ is determined by the trajectories ($p = 100$) generated and the number of parameters ($k = 9$) for each taxa ($N = p \times (k + 1)$). For each run, we compute the regional average maximum pollen emission over the US for 1 year (2015). Analyzing the value of input parameters and the model outputs, the Morris sensitivity package calculates the ratios of model output changes to the parameter variation, then computes the absolute values of mean ($\mu^*$) and standard deviation ($\sigma$) for each input parameter. The magnitude of $\mu^*$ shows the overall influence on the model output, where a large $\mu^*$ indicates the input parameters important in determining the model output. $\sigma$ is used to detect the non-linearity and interaction of the input parameters, where a large $\sigma$ suggests the parameter has a nonlinear effect on the model or this parameter is interacting with other parameters.

For each taxon, we computed the ranks of the Morris indices ($\mu^*$ and $\sigma$) for the input parameters and evaluated their relative importance (Supplementary Table 2). The four highest-ranked variables indicate a larger overall influence on model output. Generally, the top four ranked parameters of $\mu^*$ and $\sigma$ are similar between taxa, although with slightly different orders. Overall, the production-related parameters have the highest $\mu^*$ and $\sigma$ for most taxa, where the annual pollen production ($P_{annual}$) and normalization parameter($P_{norm}$) are the two most important factors. This rank is expected as these two factors directly impact the magnitude of simulated pollen emission. Phenology factors (e.g., $m_{sDOY}$, $b_{sDOY}$, $m_{eDOY}$, and $b_{eDOY}$) control the timing and variation of daily pollen emissions and are relatively less influential on the simulated pollen. However, for the taxa that exhibit a strong temperature dependence on the pollen season duration (e.g., *Alnus, Platanus, Populus*, late-flowering *Ulmus*, C4 grass), the pollen phenology factors are more important for the simulated maximum pollen emission (Supplementary Table 2).

**Reporting summary**. Further information on research design is available in the Nature Research Reporting Summary linked to this article.

## Data availability

Raw data of the simulated historical and future daily pollen emission for 15 CMIP6 models generated in this study are available from www.deepblue.lib.umich.edu under access code https://doi.org/10.7302/1s0g-b468[75]. The processed data used to produce all figures are available at https://doi.org/10.7302/628t-r416[76]. Source data are provided with this paper.

## Code availability

The source code of the pollen emission model (PECM) used to produce pollen data is available in the Github repository (https://github.com/steiner-lab/pecm) under access code https://doi.org/10.5281/zenodo.5874177[77].

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

## Acknowledgements

This work was supported by the National Science Foundation grant AGS-1821173 to ALS. Climate data are provided by the CMIP6 archive (https://esgf-node.llnl.gov/search/cmip6/; see detailed model information in "Methods"). We gratefully acknowledge the AAAAI United States pollen count data in the development of the PECM model. We thank Manish Verma at the University of Michigan Consulting for Statistics, Computing and Analytics Research (CSCAR) for guidance on model uncertainty analysis.

## Author contributions

Y.Z. performed all of the research included in the paper. A.S. designed and discussed the outline of the research with Y.Z. and advised on the methodology and the visualizations produced by this work. Designing and writing of the paper was completed by Y.Z. A.S. contributed to the improvement of the writing clarity and grammar of the manuscript.

## Competing interests

The authors declare no competing interests.
