## [Peer Review File · Nature Communications]

Reviewers' Comments:

Reviewer #1:

Remarks to the Author:

General comments

In this study, the authors parameterize a pollen emission model for the US and drive it with future climate projections to examine pollen emissions under climate scenarios. Pollen is a major airborne pollutant and a broad body of research has connected pollen emissions to climate, thus this study's future projections fill an incredibly important gap in the literature. Overall, the study is quite well-written and makes important contributions in many areas, but I have two very large concerns that need to be addressed.

First and most importantly, I do not understand why the pollen emissions model only adjusts phenology as a function of temperature and precipitation, but not the maximum daily emissions (i.e. the peak of the gaussian distribution that it fits). This seems both incorrect in that it is inconsistent with recent large-scale analyses that observe both phenology and maximum daily emissions changing in decoupled ways and purely unnecessary from an experimental design point of view. Surely a more realistic and rigorous model could be fit that allows the phenology and maximum emissions to be fit from the data as a function of climate variables. As it is, much of the changes in maximum daily emissions seem to be a pure artifact of the phenology changes driving the statistical model.

Second, the authors need to highlight and emphasize that there is a massive limitation and uncertainty in these projections in the form of shifts of species' distributions and biomass levels, all of which are climate sensitive, widely documented as occurring across the US, and not treated here. I agree that this is a very hard thing to model and no models are great at this right now (there are a few potentially simple ways this might be analyzed), but readers need to be clearly alerted that these changes in vegetation are already occurring and will have enormous impacts on pollen emissions in the US. In fact, changes in vegetation distributions and biomass might have much larger effects than the temperature-driven phenology effects fit in the current models. This isn't to say that the current models aren't worthwhile and informative (assuming the above maximum daily issues can be fixed), but it does highlight that two of the potentially largest uncertainties that the pollen projection / plant ecology and physiology fields need to grapple with are 1) CO₂ effects and 2) vegetation shifts.

Specific comments

L31: This decrease seems at odds with the conclusions below in L33 and L35. I realize there is a lot of complexity in space and time, but the net effect is that readers are a bit unclear about the overall picture. Perhaps removing this specific set of numbers from the abstract might make it more understandable?

L52: There is also so emerging evidence that high pollen levels may increase vulnerability to respiratory viruses (see a recent paper in PNAS about this)

L113-115: Why is this annual emission level imposed as a constant? This would then indicate that the decreases in daily pollen emissions are not necessarily a robust result, but instead are the output of decisions in experimental design and analysis. If that's true, those changes should definitely be removed from the abstract.

L153: Yet daily maximum pollen levels can be decoupled from seasonal lengths (i.e. it's possible for PDF of the pollen over the year to get both broader AND taller in terms of pollen concentrations on the y-axis). Does this model allow that estimation?

L166-169: Yet this is an outcome entirely of only fitting a fixed-height gaussian distribution, correct? Why not fit a 3 parameter gaussian distribution where the height (i.e. maximum daily pollen levels) can vary as a function of climate? Recent research (e.g. Anderegg et al. 2021 PNAS) has documented both increases in pollen season length (seen here in these models) AND increases in daily maximum pollen levels, so this seems like it's important to model both.

L198-199: But this is only the short-term effects of precipitation. Longer-term effects (e.g. weeks to months) include higher precipitation increasing plant biomass and pollen production, especially in water-limited regions. That needs to be clearly acknowledged somewhere.

L277-278: Please define phenology more precisely – how was that quantified?

L311-317: This is a very important sensitivity test. I am glad the authors did this.

L333-339: To me, this seems like perhaps the single most important limitation and largest uncertainty in the study – future changes in species' distributions and biomass. If it's not too much work, I wonder if a quick sensitivity analysis of looking at the projected changes in vegetation biomass in several CMIP6 models in the 2080-2100 period and adjusting the historical values based on those % changes might be informative. At the very least, this caveat needs to be more clearly highlighted in the paper discussion.

Reviewer #2:

Remarks to the Author:

This is a well thought out study that explains of the effect of specific drivers on pollen emissions and estimates future emissions dependent on temperature, precipitation, and CO₂ changes. The processes that drive emissions were explained well and the results novel. The manuscript is well-written and concise.

Line 60: Temperature impacts more than just spring frost-free days. Some plants are impacted by chilling requirements during the winter. Warmer winters can reduce chilling and delay flowering for some trees. Although, I do not know whether this will affect the timing of emissions of pollen for trees with allergenic pollen. Perhaps you can do a literature search on this and provide a summary sentence and talk about the ramifications of this with increasing temperatures with climate change and the PECM model.

Line 69-72: You mention that radiation, wind speed and humidity (line 60) also affect atmospheric pollen. Can you give a reason why you analyzed only temperature and precipitation and not the other variables as well?

Line 113-115: "...this asymmetric increase in the sDOY and longer duration flattens the pollen emission curve, decreasing daily maximum pollen emissions." I think the assumption here is that the total annual pollen emission does not change with a change in season duration and that is why daily maximum emissions decrease. Can you provide some background and/or justification for this assumption?

Figure 1: I suggest using a different color scheme for (h) because I usually associate a blue T change with a decrease in temperature. It's a little difficult to see the 5 subregions in Fig 1d.

Line 128-131: There is a discrepancy here: you say that category 3 plants are controlled by light, however line 132 and figure S3e shows that there is a temperature dependence for sDOY.

Line 141: The text says: "8 different deciduous genera in this category (Fig. 2; circles)". However, Fig 2 shows 11 circles.

Line 175: Is GFDL-CM4 representative of all the CMIP6 models in the way Quercus and Acer interact?

Line 197-207: You've shown that increase precipitation rate decreases the pollen emission in NE, SE and PNW. Fig S1f shows that there is decrease in precipitation rate over CA and the southern part of MT. Does decrease in precipitation rate also mean an increase in the pollen emission?

Fig S5e Could you extend the x-axis to include the entire curve?

Line 246-255: Do you have estimates of uncertainty for the pollen emissions and how much each factor contributes to the uncertainty? You have stated that the γ_{CO2} is uncertain. In CMIP models, the uncertainty in projections of precipitation is much greater than temperature.

Figure S4 caption: typo "imulation"

Reviewer #3:

Remarks to the Author:

My mandate was to review the climate change modelling. So my comments are restricted to this aspect of the manuscript. While I really like the research idea for this paper, I have some concerns.

My largest concern is that the authors estimate the effects of climate change on pollen emissions without considering the direct effect that climate change will have on plant distributions and community compositions. Since pollen production will be affected in space and time by the redistributions of plants under near future climate change, changes in plant community composition need to be accounted for directly in this analysis. A way to do this would be to use a dynamic vegetation model (validated on current-day distributions) to project future distribution shifts and their effects on plant community composition. This is fairly straight forward and might only need to be done for the 4 broad plant functional types.

The authors need to do a much more thorough sensitivity analysis of the Pollen Emissions Climate Model. At the moment it is restricted to assessing scenarios of CO₂. To do this I suggest that the authors use Latin hypercube sampling to ensure a good coverage of multi-dimensional space. There are plenty of papers on this technique and at least one R package. A similar approach could be used to capture uncertainty in the results.

More detail is needed on how the authors downscaled the climate data to a common grid. The link to the table in the supplementary material does not include spatial resolution for a number of models.

I am guessing that the authors calculate pollen emissions for each species for each model and then calculate an evenly weighted multi-model average? The alternative approach is to calculate a multi-model average of the daily climate projections and then pass them through the pollen model. Both approaches are used commonly, however, the reader needs to know which approach was used here.

Results in Figure 4 would be more robust if they showed a multi-model average climate projection rather than results from a single model.

Manuscript NCOMMS-21-14254-T
Response to reviewers

Dear reviewers,

Thank you for reviewing the manuscript “Projected climate-driven changes in pollen emission season length and magnitude over the continental United States” for publication in *Nature Communications*. We appreciate the time and effort taken to review our paper and provide insightful and constructive comments. We carefully considered and addressed the comments and suggestions provided by the reviewers. We appreciate all the constructive comments, as it helped us to redesign and improve the current pollen emission model.

Our point-by-point response to the reviewers’ comments and concerns is provided below, along with a tracked changed version of the manuscript that highlights all changes. We also produce a final version of the revised manuscript, with all line numbers included below referring to the final untracked version.

Response to reviewer 1:

Comment:

In this study, the authors parameterize a pollen emission model for the US and drive it with future climate projections to examine pollen emissions under climate scenarios. Pollen is a major airborne pollutant and a broad body of research has connected pollen emissions to climate, thus this study's future projections fill an incredibly important gap in the literature. Overall, the study is quite well-written and makes important contributions in many areas, but I have two very large concerns that need to be addressed.

Response:

Thank you for the positive feedback. The two concerns are addressed below.

Comment:

First and most importantly, I do not understand why the pollen emissions model only adjusts phenology as a function of temperature and precipitation, but not the maximum daily emissions (i.e. the peak of the gaussian distribution that is fit). This seems both incorrect in that it is inconsistent with recent large-scale analyses that observe both phenology and maximum daily emissions changing in decoupled ways and purely unnecessary from an experimental design point of view. Surely a more realistic and rigorous model could be fit that allows the phenology and maximum emissions to be fit from the data as a function of climate variables. As it is, much of the changes in maximum daily emissions seem to be a pure artifact of the phenology changes driving the statistical model.

Response:

Thank you for pointing out this shortcoming. We agree with the reviewer that only considering plant phenology as a function of climate change does not capture the full scope of how climate may influence pollen production.

To address this concern, we analyze the relationships between observed annual total pollen counts and other climate drivers. We examined both temperature and precipitation, as these are readily available output from climate models and important drivers of ecosystem dynamics (Wu et al., 2011). Relationships between observed pollen count and precipitation were not evident, despite trying numerous variables such as average daily precipitation, annual total precipitation, seasonal precipitation, and annual precipitation variation (see a visualization of this in the response below on Page 10, Figure R1). For temperature, however, statistically significant relationships were evident and are also supported in the literature (Ziello et al., 2012, Anderegg et al., 2020). Therefore, we included a temperature-dependent component to the pollen production factor. We alter the annual total pollen production instead of maximum daily pollen production as this fits with our existing model framework. Additionally, because NAB pollen measurements are frequently collected for part of the year or certain days of a week (Anderegg et al., 2020), the daily metrics (e.g., maximum daily pollen emission) have larger uncertainties.

Correlations between temperature and pollen count for each plant taxa used in the model regression are shown in the Supplementary material Fig. 3 and summarized in Table S1:

Figure S3 Relationships between annual total pollen counts and temperature.

Linear regressions for log-transformed accumulated annual total pollen counts versus previous-year annual average temperature (PYAAT, °C) for all taxa. Each point signifies one station per year from 2003 to 2010, with N as the total number of observations utilized in the regression. Regression statistics are summarized in Table S1.

The addition of a temperature-dependent production term is now described in the revised model methodology:

Line 408-432:

Methods:

“ In PECM1.0, taxa-dependent pollen production values by taxa (P_{annual} ; grains tree⁻¹ converted to grains m⁻²) are derived from limited literature field surveys^{47,54–62} and are an important input (see Model Sensitivity Analysis). P_{annual} values are updated from Wozniak et al.⁴¹ to include new literature (Supplementary Table 1). However in PECM1.0, P_{annual} was held constant for each taxon and does not account for potential interannual variation of pollen production with time. A few studies have examined the interannual variability of pollen count^{63,64}, but a complication of using atmospheric pollen count to determine the interannual variability of production is that many meteorological factors influence the count (e.g., wind, precipitation, and other meteorological conditions such as boundary layer height). Therefore we scale the literature production factors (P_{annual}) to a temperature-dependent annual production factor (pf_{annual}) with the observed linear relationship between log-transformed observed annual total pollen

counts and previous-year annual average temperature (PYAAT) for each taxa (Equation (5)) normalized to the temperature-dependent production (P_{norm}):

$$pf_{annual} = \frac{\exp(m_{prod} \times PYAAT + b_{prod})}{P_{norm}} \times P_{annual} \quad (5)$$

m_{prod} and b_{prod} are the slope and intercept of the linear regression, respectively, of observed pollen count and temperature and vary for each taxon (Supplementary Table 1). P_{norm} is the pollen count at the historical average temperature over the US. One exception is for the genus *Alnus*, which has a limited spatial range in the Pacific Northwest (PNW) and we use the historical average temperature in the PNW.

Nine of the 15 total PYAAT-pollen count relationships are statistically significant ($P < 0.05$, Supplementary Fig. 3), suggesting that this regression captures the influence of prior year temperature on the subsequent year's pollen production. While there are likely many other factors that influence pollen production (e.g., moisture resources, soil nitrogen, etc.), this provides a simple method to account for year over year pollen production. ...”

Overall, this adjustment to the production factors yields similar values to the prior production constants for the historical period (most taxa have less than 20% of changes).

Comment:

Second, the authors need to highlight and emphasize that there is a massive limitation and uncertainty in these projections in the form of shifts of species' distributions and biomass levels, all of which are climate sensitive, widely documented as occurring across the US, and not treated here. I agree that this is a very hard thing to model and no models are great at this right now (there are a few potentially simple ways this might be analyzed), but readers need to be clearly alerted that these changes in vegetation are already occurring and will have enormous impacts on pollen emissions in the US. In fact, changes in vegetation distributions and biomass might have much larger effects than the temperature-driven phenology effects fit in the current models. This isn't to say that the current models aren't worthwhile and informative (assuming the above maximum daily issues can be fixed), but it does highlight that two of the potentially largest uncertainties that the pollen projection / plant ecology and physiology fields need to grapple with are 1) CO2 effects and 2) vegetation shifts.

Response:

Thank you for this comment, and we agree with the reviewer that future plant biomass and distribution change will have large effects on the pollen emission.

To address this concern, we added a sensitivity test to address the effects of future plant distribution shift on pollen maximum emission. PECCM can calculate pollen emissions based on two different types of land cover: taxa-based land cover (as used in this study) or plant functional type (PFT)-based land cover (Wozniak and Steiner, 2017). Generally, the taxa-based model evaluates better than the PFT model when compared against observations (Wozniak and Steiner, 2017), therefore we used the taxa-based model in our study. However, we could not find gridded maps of taxa-level spatial distribution shifts due to climate but there were future projections of PFT-based land cover changes (Chen et al., 2020). Therefore we used the PFT version of the model for this sensitivity test. These results and discussion are now included in the manuscript on line 282-308 and Fig. 4p-t (included below):

Line 282-308:

Main

“Future climate change and anthropogenic impacts are likely to shift the spatial distribution of plant communities²⁷ and therefore impact pollen emissions. Because gridded taxa-specific land cover change data is not available, we test the impact of land cover change using projections of plant functional types (PFTs) from GCAM-Demeter land use dataset⁴⁹ (Supplementary Fig. 6). PECM1.0 can estimate pollen emissions based on PFT with greater uncertainties than the taxa-specific method employed for the previous simulations⁴¹. We simulate the future maximum pollen emission with PFT-based pollen emission model using both the historical and future land cover (see Methods) (Supplementary Fig. 7). The difference between the two PFT simulations indicates the impact of future land cover change on maximum pollen emissions over the US, noting that the pf_{annual} vary between the PFT and taxa-based models (Supplementary Table 1).

Future PFT changes projected by the Global Change Analysis Model (GCAM)⁴⁹ simulate an increase in tree coverage in the Central US and Mississippi River Valley at the expense of crop, and some decrease in tree coverage at high altitudes of the Rockies and the Pacific Northwest (Supplementary Fig. 6). Compared to tree PFTs, changes to grassland are relatively small (-40%) and occur in smaller patches (Supplementary Fig. 6c). Ragweed coverage is based on urban and cropland, and the future reduction of the cropland due to the expansion of grasses or trees⁴⁹ drives large decreases (80%) of potential ragweed land cover over the eastern US (Supplementary Fig. 6d). For the two dominant US tree types (DBL and ENL), the regional pollen maximum emission increases up to 6% in SE and CA (California) while decreasing up to 7% in the MT and PNW regions (Fig. 4p, q, t). Future changes to grass cover are relatively small, leading to an increase in grass pollen emissions of about 4-10% (Fig. 4t). Ragweed emissions have the largest pollen emission maxima decreases over NE and SE (up to 32%; Fig. 4t) because of cropland reduction, however we note large uncertainties in the spatial distribution of ragweed. Compared to emission changes due to climate or CO₂ effects, the maximum pollen emission changes due to land cover changes at the regional scale are relatively small (-32% to 6%).”

Overall, the sensitivity test indicates that the plant distribution is another factor that will likely impact future pollen emissions, although the magnitude is less than that of climate or CO₂.

The impact of future climate on plant biomass is slightly more challenging to address. However, we include an estimate of this effect through the change in the total pollen production. We agree that climate change will impact the plant growth and biomass, and thereby alter the total pollen production. By including the new method that simulates the total pollen production as a function of temperature change (and CO₂ increase in the sensitivity test), we are likely including the corresponding change of plant biomass. We specifically highlight this connection in the revised manuscript:

Line 159-161:

Main:

“In addition to altering phenology, warmer temperatures will also impact vegetation physiology, either facilitating or restraining growth and impacting plant biomass. These changes have the potential to increase or decrease the annual total pollen production^{37,47} (pf_{annual} ; Equation (5)).”

Fig. 4:

Fig.4: Historical and future changes of maximum daily pollen emissions ($E_{pol,max}$)

a-e, Multi-model historical average (1995-2014) maximum daily pollen emission flux ($E_{pol,max}$) over the United States ($grains\ m^{-2}\ d^{-1}$). **f-j**, Projected multi-model average future $E_{pol,max}$ change (%) at the end of century (2081-2100) for SSP 585, with the effects of temperature (T) and precipitation (pr) only, and **(k-o)** Projected future $E_{pol,max}$ change (%) due to temperature, precipitation, and CO₂. Panels **a-o** use the taxa-based pollen emission model (PECM) driven by meteorology input data from each CMIP6 model to calculate the multi-model average. **(p-t)** PFT-based model $E_{pol,max}$ change (%) with land cover change (LCC) effects only. The simulation is conducted using PFT-based pollen emission model (PECM) with historical (2015) and future (2100) PFT land cover and driven by the multi-model average meteorology input data. Columns represent different PFTs: DBL (**a, f, k**), ENL (**b, g, l**), GRA (**c, h, m**), RAG (**d, i, n**). Bar charts (**e, j, o, t**) show the spatial averages in 5 subregions (Fig. 3g)) with error bars representing the standard deviation from the average of multiple models in each region (**e, j, o**).

Comment:

L31: This decrease seems at odds with the conclusions below in L33 and L35. I realize there is a lot of complexity in space and time, but the net effect is that readers are a bit unclear about the overall picture. Perhaps remove this specific set of numbers from the abstract might make it more understandable?

Response:

Thank you for noting this point in the abstract. We agree with the reviewer that this decrease (caused by the extended pollen season duration and the fixed pollen production) is confusing, and this conclusion is revised after revising the pollen production factor as suggested above. In the new simulations, the maximum daily pollen emissions increase over most of the US, with some decreases over some regions because of the competing effects of pollen season extension and pollen production change. However, the annual total pollen emission increases over all the regions. We have updated this text in the abstract:

Line 30-32:

“Temperature and precipitation alter daily pollen emission maxima by -35 to 40% and increase the annual total pollen emission by 16-40% due to changes in phenology and temperature-driven pollen production.”

Comment:

L52: There is also so emerging evidence that high pollen levels may increase vulnerability to respiratory viruses (see a recent paper in PNAS about this)

Response:

Thank you for pointing it out. We were aware of this connection, but still feel that the respiratory virus connection needs more rigorous data analysis and have not included this in the revised manuscript.

Comment:

L113-115: Why is this annual emission level imposed as a constant? This would then indicate that the decreases in daily pollen emissions are not necessarily a robust result, but instead are the output of decisions in experimental design and analysis. If that's true, those changes should definitely be removed from the abstract.

Response:

We agree with the reviewer's assessment. As noted in the previous response above, we have changed the fixed annual pollen production into one that is dynamic and temperature-dependent. As a result, we believe that we have considered all possible factors that we can given the limited literature and have quantified the simulated changes in the abstract.

Comment:

L153: Yet daily maximum pollen levels can be decoupled from seasonal lengths (i.e. it's possible for PDF of the pollen over the year to get both broader AND taller in terms of pollen concentrations on the y-axis). Does this model allow that estimation?

Response:

In the original version that we submitted, the answer to the reviewer's question would be no. However, we have revised the production term to depend on temperature, thereby allowing interannual variability in the pollen production (See the Methods section and above responses). It is possible in the revised simulations to make the Gaussian distribution of pollen emission both taller and broader. In the new model version, many taxa have shown both taller and broader PDF

of pollen emission, as shown in the conceptual figures of pink curves of category 1 and category 3 for Fig. 2a,c in the revised manuscript.

Fig.2: Three categories of pollen emission changes

a-c, Conceptual schematic for three categories of pollen emission changes, comparing the historical (black) and future (colors) daily pollen emission. Blue lines indicate the future pollen emission change with only temperature-dependent phenology. Pink and purple curves represent future pollen emission change with phenology change and temperature-dependent changes in the annual production factor (pf_{annual}), which can increase (pink) or decrease (purple) production.

However, due to the different responses of different taxa to temperature change (see above on page 3, Supplementary Fig.3), not all taxa will have both broader and taller PDF in the future. For the taxa with lower pollen production in the future (e.g., purple curves in category 1 and 2 of Fig. 2a, b), pollen maximum daily emission decreases and lead to shorter and broader PDF.

We updated the text in the revised manuscript to describe this phenomenon:

Line 162-174:

“Limited observational data suggests that many taxa are projected to have higher pollen production with warmer temperatures (e.g., *Alnus*, *Morus*, *Platanus*, *Quercus*, spring and late-flowering *Ulmus*, Cupresseceae, *Ambrosia*, C3 and C4 Poaceae³), while some taxa will have lower pollen production (e.g., *Acer*, *Betula*, *Fraxinus*, *Populus*, Pinaceae) (Supplementary Table 1 and Fig. 3)^{37,48}. The variations in pf_{annual} in conjunction with phenology can influence the maximum daily pollen emission ($E_{pol,max}$; $grains\ m^{-2}d^{-1}$), which is a potential health exposure metric. Increasing pf_{annual} under future climate typically increases the $E_{pol,max}$ and counteracts the lengthening season duration for Category 1 and 3 vegetation (Fig. 2a, c, respectively), especially for taxa with a strong dependence on temperature (e.g., *Alnus*, *Platanus*, *Quercus*, late-flowering *Ulmus*, Cupresseceae, *Ambrosia*; Supplementary Fig. 3). In contrast,

decreasing pf_{annual} will further decrease the $E_{pol,max}$ (Fig. 2a and b), with greater decreases for taxa projected to have longer durations in the future (*Acer*, *Fraxinus*, *Populus*, Pinaceae; Fig. 2a).”

Comment:

L166-169: Yet this is an outcome entirely of only fitting a fixed-height gaussian distribution, correct? Why not fit a 3 parameter gaussian distribution where the height (i.e. maximum daily pollen levels) can vary as a function of climate? Recent research (e.g. Anderegg et al. 2021 PNAS) has documented both increases in pollen season length (seen here in these models) AND increases in daily maximum pollen levels, so this seems like it's important to model both.

Response:

Thank you for the suggestions. As noted in the above responses, we edited the model accordingly to allow for a change in production from year to year. However, instead of altering the height as suggested, we changed the full Gaussian distribution via the total production value, as there is some evidence that temperature can directly impact total pollen production by affecting the plant biomass (Fumanal et al., 2007). The changed total pollen production parameterization increases in the maximum pollen emission over most of the regions because temperatures are generally warming over the continental US. However, for some taxa like *Betula*, warming temperatures do not lead to an increase in production and CO₂ plays a more important role than temperature as mentioned in Ziello et al., 2012. Therefore, after adding CO₂ effects, maximum pollen emission increase over all of the region for all of the taxa. This trend corresponds to the results presented in Anderegg et al. (2021).

Comment:

L198-199: But this is only the short-term effects of precipitation. Longer-term effects (e.g. weeks to months) include higher precipitation increasing plant biomass and pollen production, especially in water-limited regions. That needs to be clearly acknowledged somewhere.

Response:

We appreciate this point.

We have evaluated the longer-term effects of precipitation on pollen production. However, due to the complexity of precipitation effects on pollen production and atmospheric transport, we could not find any significant correlations between longer-term precipitation and pollen production as shown in Fig. R1 below. Only 6 in 15 taxa show statistically significant correlations ($P < 0.05$) and the response is mixed in sign.

Figure R1: Linear relationship between log transformed observed annual total pollen counts and annual total precipitation (mm).

Linear regressions for log transformed accumulated annual total pollen counts versus annual accumulated precipitation (Annual total pr, mm) for all taxa. Each point signifies one station per year from 2003 to 2010, with N as the total number of observations in regression.

Therefore, we only included the precipitation short-term effects in the model and discussed the dual effects of precipitation in the revised manuscript:

Line 444-449:

Methods:

“Precipitation exerts a dual effect on pollen emission¹¹. High precipitation will remove emitted pollen grains from the atmosphere and can prevent pollen release, while low annual precipitation may limit the plant growth and may restrain the pollen production. However, due to the complexity of precipitation effects, we did not find robust correlations between precipitation and pollen production at longer time scales. Therefore, we only include the precipitation scavenging effects in this study through a precipitation factor (γ_{precip}).”

Comment:

L277-278: Please define phenology more precisely – how was that quantified?

Response:

Our use of the term “phenology” refers to the pollen season timing, or the start day of the pollen season (start day of year; sDOY) and end date of pollen season (end day of year; eDOY). We have clarified this term in the manuscript.

Line 379-381:

Methods:

“It predicts pollen emission for a broad range of taxa at a large geographic scale (25 km resolution in this study), and it is able to capture up to 57% of the variance of pollen season timing of the sDOY and eDOY.”

Comment:

L311-317: This is a very important sensitivity test. I am glad the authors did this.

Response:

Thank you very much!

Comment:

L333-339: To me, this seems like perhaps the single most important limitation and largest uncertainty in the study – future changes in species’ distributions and biomass. If it’s not too much work, I wonder if a quick sensitivity analysis of looking at the projected changes in vegetation biomass in several CMIP6 models in the 2080-2100 period and adjusting the historical values based on those % changes might be informative. At the very least, this caveat needs to be more clearly highlighted in the paper discussion.

Response:

We agree that this test would improve the manuscript. We conducted a sensitivity test for the future species’ distribution change impacts on pollen emissions (please see the previous response on page 4-6 and Fig. 4p-t). After careful consideration, we didn’t add a separate sensitivity test for plant biomass change because the pollen production change with temperature was intended to capture a biomass change effect. We have included the results of the spatial distribution shift on lines 282-308 of the revised manuscript and show the results in Fig. 4.

Response to reviewer 2:

Comment:

This is a well thought out study that explains of the effect of specific drivers on pollen emissions and estimates future emissions dependent on temperature, precipitation, and CO2 changes. The processes that drive emissions were explained well and the results novel. The manuscript is well-written and concise.

Response:

Thank you for the positive comments!

Comment:

Line 60: Temperature impacts more than just spring frost-free days. Some plants are impacted by chilling requirements during the winter. Warmer winters can reduce chilling and delay flowering for some trees. Although, I do not know whether this will affect the timing of emissions of pollen for trees with allergenic pollen. Perhaps you can do a literature search on this and provide a summary sentence and talk about the ramifications of this with increasing temperatures with climate change and the PECM model.

Response:

Our start day timing of pollen is dependent on the prior year annual average temperature (PYAAT) and does not have a chilling requirement, as some prior studies have noted that the chilling phase is not the primary driver for the budburst process (Fu et al., 2012; Grundstrom et al., 2019) and note that the impact of warmer winters on the chill requirement could either increase or decrease the role of spring temperatures (Newnham et al., 2012).

But we agree with the reviewer on this point, and have noted this on lines 60-61 of the revised manuscript:

“Temperature impacts the number of winter chill hours and spring frost-free days and is”

Comment:

Line 69-72: You mention that radiation, wind speed and humidity (line 60) also affect atmospheric pollen. Can you give a reason why you analyzed only temperature and precipitation and not the other variables as well?

Response:

We thank reviewer for pointing this out.

Our prior work has identified the two factors that influence the pollen production the most: temperature and precipitation. The factors of radiation, humidity and wind speed are most important for the conditions that lead to the release of pollen to the atmosphere, and play a greater role on pollen counts which are based on atmospheric observations. Our prior work has included these meteorological effects when the pollen emission is coupled to an atmospheric model (e.g., as in Wozniak et al., 2018). However, in this work we focus on the emissions potential (or how much pollen plants produce and have the *potential* to emit). The primary reason for this is that running the coupled atmospheric model is much more computationally expensive than running the emissions model, and we did not have the resources to run the

coupled model for all emissions scenarios and models presented in the manuscript. We have clarified this point in the manuscript on lines 451-455:

Line 451-455:

“While other atmospheric conditions such as wind and humidity are linked to precipitation changes and may inversely lead to higher pollen concentration before or during the rainfall events^{65,66}, this study does not account for meteorological factors other than temperature and precipitation, as this requires coupling with an atmospheric model as in Wozniak et al.⁴¹”

Therefore, investigating the effects of these factors exceeds the scope of this study, but we will investigate their interaction with emitted pollen in our future study by coupling with WRF-Chem.

To reduce the confusion, we removed the description that radiation, wind speed and humidity can impact the pollen emission in the Line 59-60 of the revised manuscript:

“Pollen emission potential of anemophilous plants is directly correlated with meteorological conditions such as temperature and precipitation^{17,18}”

Comment:

Line 113-115: “...this asymmetric increase in the sDOY and longer duration flattens the pollen emission curve, decreasing daily maximum pollen emissions.” I think the assumption here is that the total annual pollen emission does not change with a change in season duration and that is why daily maximum emissions decrease. Can you provide some background and/or justification for this assumption?

Response:

Thank you for this comment. The reviewer is correct that in the previous manuscript we fixed the annual total pollen production. Because the literature on pollen production is extremely limited and based on field surveys, we could find no available data on the changes in pollen production over time. However, based on this comment and those from other reviewers (see page 2 and 7 of this response), we modified the emission model to allow the annual total pollen production to vary with time. The new model parameterization connects the annual total pollen production with the annual total pollen counts, making annual total pollen production change with temperature.

Line 408-432:

Methods:

“ In PECM1.0, taxa-dependent pollen production values by taxa (P_{annual} ; grains tree⁻¹ converted to grains m⁻²) are derived from limited literature field surveys^{47,54-62} and are an important input (see Model Sensitivity Analysis). P_{annual} values are updated from Wozniak et al.⁴¹ to include new literature (Supplementary Table 1). However in PECM1.0, P_{annual} was held constant for each taxon and does not account for potential interannual variation of pollen production with time. A few studies have examined the interannual variability of pollen count^{63,64}, but a complication of using atmospheric pollen count to determine the interannual variability of production is that many meteorological factors influence the count (e.g., wind, precipitation, and other meteorological conditions such as boundary layer height). Therefore we scale the literature

production factors (P_{annual}) to a temperature-dependent annual production factor (pf_{annual}) with the observed linear relationship between log-transformed observed annual total pollen counts and previous-year annual average temperature (PYAAT) for each taxa (Equation (5)) normalized to the temperature-dependent production (P_{norm}):

$$pf_{annual} = \frac{\exp(m_{prod} \times PYAAT + b_{prod})}{P_{norm}} \times P_{annual} \quad (5)$$

m_{prod} and b_{prod} are the slope and intercept of the linear regression, respectively, of observed pollen count and temperature and vary for each taxon (Supplementary Table 1). P_{norm} is the pollen count at the historical average temperature over the US. One exception is for the genus *Alnus*, which has a limited spatial range in the Pacific Northwest (PNW) and we use the historical average temperature in the PNW.

Nine of the 15 total PYAAT-pollen count relationships are statistically significant ($P < 0.05$, Supplementary Fig. 3), suggesting that this regression captures the influence of prior year temperature on the subsequent year's pollen production. While there are likely many other factors that influence pollen production (e.g., moisture resources, soil nitrogen, etc.), this provides a simple method to account for year over year pollen production. ...”

Comment:

Figure 1: I suggest using a different color scheme for (h) because I usually associate a blue T change with a decrease in temperature. It's a little difficult to see the 5 subregions in Fig 1d.

Response:

Thank you for the visualization suggestions. We have revised this figure (Fig. S1 of the revised supplementary material) as suggested with only red scale temperature changes:

We also removed the 5 subregions division from this figure and added it to Fig. 3g of the revised manuscript to enable viewing of the regions.

Comment:

Line 128-131: There is a discrepancy here: you say that category 3 plants are controlled by light, however line 132 and figure S3e shows that there is a temperature dependence for sDOY.

Response:

Sorry for the confusion. The flowering and pollen production of category 3 plants are impacted by light (known as short-day species), and they are also impacted by temperature. When we developed the model, the sDOY and eDOY regression statistics were better for correlations with temperature rather than light, and the determination of their timing is based on temperature (explained in Wozniak and Steiner, 2017). We have modified the text on lines 136-139 to make this point more clear.

lines 136-139:

“. Despite seasonal light as a driver, sDOY and eDOY for these taxa was best predicted with temperature⁴¹. Both sDOY and eDOY increasing with warmer temperatures (triangles in Fig. 1) and are projected to occur later under end-of-century conditions, with a stronger eDOY temperature dependence than sDOY resulting in a longer duration (Fig. 2c).”

Comment:

Line 141: The text says: “8 different deciduous genera in this category (Fig. 2; circles)”.

However, Fig 2 shows 11 circles.

Response:

Apologies for this error - we corrected the text accordingly. Except 8 deciduous genera, there are also 2 evergreen genera and 1 grass family in the first category.

Line 147-148:

“...the duration increases in Category 1 are 2-19 days, where the large variability is a function of the 11 different vegetation taxa in this category (Fig. 1; circles).”

Comment:

Line 175: Is GFDL-CM4 representative of all the CMIP6 models in the way *Quercus* and *Acer* interact?

Response:

Thanks for your question. The short answer is yes: because all the CMIP6 models have similar temperature trends (but slightly different magnitudes), the interaction of *Quercus* and *Acer* is similar for all 15 climate models. We selected GFDL-CM4 because it has the lowest temperature bias when compared to observations throughout the US.

However, we do realize only using one model (GFDL-CM4) to represent all CMIP6 models is confusing. Therefore, in the revised manuscript, we use the multi-model average simulation result to show the pollen season overlap.

Fig.3: Simulated regional pollen seasonal magnitude and timing in the historical and future.

a-f, 20-year average time series of daily pollen emission flux ($grains\ m^{-2}d^{-1}$) of the six dominant individual tree taxa in DBL and the total DBL emission with (solid black line) and without (dashed black line) precipitation effects. (**a-c**) Historical (1995-2014) emission and (**d-f**) end of the century (2081-2100) emissions from multi-model average simulation for SSP 585. **g**,

There are 5 geographic regions used in this study: Northeast (NE; 38-48° N and 70-100° W), Southeast (SE; 25-38° N and 70-100° W), Mountain (MT; 25-48° N and 100-116° W), California (CA; 25-40° N and 116-125° W) and Pacific Northwest (PNW; 40-48° N and 116-125° W). Three regions are selected for DBL daily pollen emission analysis: Northeast, NE (**a, d**); Mountain, MT (**b, e**); Pacific Northwest, PNW (**c, f**).

Comment:

Line 197-207: You've shown that increase precipitation rate decreases the pollen emission in NE, SE and PNW. Fig S1f shows that there is decrease in precipitation rate over CA and the southern part of MT. Does decrease in precipitation rate also mean an increase in the pollen emission?

Response:

The reviewer is correct, decreasing precipitation will reduce the scavenging effect on pollen emissions and therefore “mean an increase in the pollen emission”. For example, for MT in Fig. 3 for the climate-only simulations, precipitation removes about 17% of pollen maximum emission in the historical (Fig. 3b) and removes 14% in the future (Fig. 3e) because of the reduced precipitation rate (compare dashed and solid black lines). However, although the precipitation effect removes less pollen, the temperature effect dominates by lengthening the duration and reducing the annual production factor for *Populus*, leading to lower maximum pollen emission.

Comment:

Fig S5e Could you extend the x-axis to include the entire curve?

Response:

Thank you for pointing it out. The Fig. S5 in the previous manuscript only includes the first 180 days of a year. However, the pollen season of Cupressaceae starts from the winter (around day 320) and therefore half of the pollen emission curve is missing. In the revised supplementary Fig. S4, we extended the x-axis to a whole year.

Comment:

Line 246-255: Do you have estimates of uncertainty for the pollen emissions and how much each factor contributes to the uncertainty? You have stated that the γCO_2 is uncertain. In CMIP models, the uncertainty in projections of precipitation is much greater than temperature.

Response:

Thank you for pointing it out. In the revised manuscript, we added a new sensitivity analysis to determine the uncertainties of individual pollen parameters using the Morris Method. The result is below:

Line 547-554:

Methods:

“... the annual pollen production (P_{annual}) and normalization parameter (P_{norm}) are the two most important factors. This rank is expected as these two factors directly impact the magnitude of simulated pollen emission. Phenology factors (e.g., m_{sDOY} , b_{sDOY} , m_{eDOY} , and b_{eDOY}) control the timing and variation of daily pollen emissions and are relatively less influential on the simulated pollen. However, for the taxa that exhibit a strong temperature dependence on the

pollen season duration (e.g., *Alnus*, *Platanus*, *Populus*, late-flowering *Ulmus*, C4 grass), the pollen phenology factors are more important for the simulated maximum pollen emission (Supplementary Table 2).”

We agree with the reviewer that the precipitation projection in CMIP6 models has large uncertainties. Therefore, to constrain the uncertainties from CMIP model projections, we use multi-model average simulation result for analysis.

Comment:

Figure S4 caption: typo “imulation”

Response:

Thank you, the manuscript is corrected accordingly.

Response to reviewer 3:

Comment:

My mandate was to review the climate change modelling. So my comments are restricted to this aspect of the manuscript. While I really like the research idea for this paper, I have some concerns.

Response:

Thank you for the kind comments. The concerns are addressed below.

Comment:

My largest concern is that the authors estimate the effects of climate change on pollen emissions without considering the direct effect that climate change will have on plant distributions and community compositions. Since pollen production will be affected in space and time by the redistributions of plants under near future climate change, changes in plant community composition need to be accounted for directly in this analysis. A way to do this would be to use a dynamic vegetation model (validated on current-day distributions) to project future distribution shifts and their effects on plant community composition. This is fairly straight forward and might only need to be done for the 4 broad plant functional types.

Response:

We agree with this suggestion. We added a sensitivity test to evaluate the effects of future plant distribution change on pollen emission. As suggested by reviewer, we use projected plant distribution data of 4 plant functional types from GCAM-Demeter land use dataset (Chen et al., 2020) to drive the PFT-version of PECM, and compare the maximum pollen emission change with historical (2015) and future (2100) land cover. The method and the results are shown below.

Fig. 4p-t:

Fig.4: Historical and future changes of maximum daily pollen emissions ($E_{pol,max}$)

a-e, Multi-model historical average (1995-2014) maximum daily pollen emission flux ($E_{pol,max}$) over the United States ($grains\ m^{-2}\ d^{-1}$). **f-j**, Projected multi-model average future $E_{pol,max}$ change (%) at the end of century (2081-2100) for SSP 585, with the effects of temperature (T) and precipitation (pr) only, and **k-o** Projected future $E_{pol,max}$ change (%) due to temperature, precipitation, and CO₂. Panels **a-o** use the taxa-based pollen emission model (PECM) driven by meteorology input data from each CMIP6 model to calculate the multi-model average. **(p-t)** PFT-based model $E_{pol,max}$ change (%) with land cover change (LLC) effects only. The simulation is conducted using PFT-based pollen emission model (PECM) with historical (2015) and future (2100) PFT land cover and driven by the multi-model average meteorology input data. Columns represent different PFTs: DBL (**a, f, k**), ENL (**b, g, l**), GRA (**c, h, m**), RAG (**d, i, n**). Bar charts (**e, j, o, t**) show the spatial averages in 5 subregions (Fig. 3g) with error bars representing the standard deviation from the average of multiple models in each region (**e, j, o**).

Line 486-487, 490-493:

Methods:

“...we include a sensitivity test of future land cover change effects on pollen emission using the PFT-based PECM model⁴¹... For the sensitivity test, we simulated the future maximum daily pollen emission both using historical (2015) and future (2100) PFT land cover data from GCAM-Demeter land use dataset⁴⁹. Compared to the taxa-based model, the PFT version of model extends to all of North America and simulates higher ENL pollen emissions over the US (Supplementary Fig. 8).”

Line 282-308:

Main:

“Future climate change and anthropogenic impacts are likely to shift the spatial distribution of plant communities²⁷ and therefore impact pollen emissions. Because gridded taxa-specific land cover change data is not available, we test the impact of land cover change using projections of plant functional types (PFTs) from GCAM-Demeter land use dataset⁴⁹ (Supplementary Fig. 6). PECM1.0 can estimate pollen emissions based on PFT with greater uncertainties than the taxa-specific method employed for the previous simulations⁴¹. We simulate the future maximum pollen emission with PFT-based pollen emission model using both the historical and future land cover (see Methods) (Supplementary Fig. 7). The difference between the two PFT simulations indicates the impact of future land cover change on maximum pollen emissions over the US, noting that the pf_{annual} vary between the PFT and taxa-based models (Supplementary Table 1).

Future PFT changes projected by the Global Change Analysis Model (GCAM)⁴⁹ simulate an increase in tree coverage in the Central US and Mississippi River Valley at the expense of crop, and some decrease in tree coverage at high altitudes of the Rockies and the Pacific Northwest (Supplementary Fig. 6). Compared to tree PFTs, changes to grassland are relatively small (-40%) and occur in smaller patches (Supplementary Fig. 6c). Ragweed coverage is based on urban and cropland, and the future reduction of the cropland due to the expansion of grasses or trees⁴⁹ drives large decreases (80%) of potential ragweed land cover over the eastern US (Supplementary Fig. 6d). For the two dominant US tree types (DBL and ENL), the regional pollen maximum emission increases up to 6% in SE and CA (California) while decreasing up to

7% in the MT and PNW regions (Fig. 4p, q, t). Future changes to grass cover are relatively small, leading to an increase in grass pollen emissions of about 4-10% (Fig. 4t). Ragweed emissions have the largest pollen emission maxima decreases over NE and SE (up to 32%; Fig. 4t) because of cropland reduction, however we note large uncertainties in the spatial distribution of ragweed. Compared to emission changes due to climate or CO₂ effects, the maximum pollen emission changes due to land cover changes at the regional scale are relatively small (-32% to 6%).”

Comment:

The authors need to do a much more thorough sensitivity analysis of the Pollen Emissions Climate Model. At the moment it is restricted to assessing scenarios of CO₂. To do this I suggest that the authors use Latin hypercube sampling to ensure a good coverage of multi-dimensional space. There are plenty of papers on this technique and at least one R package. A similar approach could be used to capture uncertainty in the results.

Response:

Thank you for the constructive comment and we agree that a sensitivity analysis would enhance the manuscript. After careful consideration, we used the sampling technique from the Morris method (Morris, 1991) instead of Latin hypercube sampling as suggested by the reviewer, because some of our parameters are not independent (e.g., the slope and intercept of regressions). The Morris method is also known as one-step-at-a-time approach, can efficiently investigate the importance of model input parameters in a multi-dimensional space. We use the Morris method to conduct the model sensitivity analysis and the description of method and the results are shown in the revised manuscript:

Line 517-554:

“Model sensitivity analysis

To evaluate the uncertainties of the PECM model, we conducted a sensitivity analysis using the Morris method⁷⁰. The Morris method is a “one-at-a-time” approach, allowing a computationally efficient uncertainty evaluation for a large number of model parameters. Nine parameters for each of the fifteen taxa used in the model are studied in this analysis (Supplementary Table 4). The uncertainty ranges of each parameter are determined by literature values (P_{annual}) or computed by the 95% confidence level (the linear regression slope (m) and intercept (b) used to calculate start date (m_{sDOY} , b_{sDOY}) and end date (m_{eDOY} , b_{eDOY}) of pollen season, and the pollen production (m_{prod} , b_{prod})). Because the normalized pollen production (P_{norm}) is calculated using the linear regression with m_{prod} and b_{prod} , its uncertainty range is also determined by the range of m_{prod} and b_{prod} . For the phenological Gaussian width a , the maximum and minimum value is obtained by ± 0.2 of the original value (3).

Using the method of Morris from the Sensitivity Analysis Library (SALib) in Python (<https://salib.readthedocs.io/en/latest/>), we conducted 1000 (N) model runs for each taxon, where the N is determined by the trajectories ($p=100$) generated and the number of parameters ($k=9$) for each taxa ($N = p \times (k + 1)$). For each run, we compute the regional average maximum pollen emission over the US for one year (2015). Analyzing the value of input parameters and the model outputs, the Morris sensitivity package calculates the ratios of model output changes to the parameter variation, then computes the absolute values of mean (μ^*) and standard deviation (σ) for each input parameter. The magnitude of μ^* shows the overall influence on the model

output, where a large μ^* indicates the input parameters important in determining the model output. σ is used to detect the non-linearity and interaction of the input parameters, where a large σ suggests the parameter has a non-linear effect on the model or this parameter is interacting with other parameters.

For each taxon, we computed the ranks of the Morris indices (μ^* and σ) for the input parameters and evaluated their relative importance (Supplementary Table 2). The 4 highest ranked variables indicate a larger overall influence on model output. Generally, the top 4 ranked parameters of μ^* and σ are similar between taxa, although with slightly different orders. Overall, the production-related parameters have the highest μ^* and σ for most taxa, where the annual pollen production (P_{annual}) and normalization parameter (P_{norm}) are the two most important factors. This rank is expected as these two factors directly impact the magnitude of simulated pollen emission. Phenology factors (e.g., m_{sDOY} , b_{sDOY} , m_{eDOY} , and b_{eDOY}) control the timing and variation of daily pollen emissions and are relatively less influential on the simulated pollen. However, for the taxa that exhibit a strong temperature dependence on the pollen season duration (e.g., *Alnus*, *Platanus*, *Populus*, late-flowering *Ulmus*, C4 grass), the pollen phenology factors are more important for the simulated maximum pollen emission (Supplementary Table 2).”

Supplementary Table 2:

Morris Index	parameters	Taxa														
		ACER	ALDR	BETU	CUPR	FRAX	MORU	PINU	PLAT	POPU	QUER	ULNU	ULN2	GRC3	GRC4	AMBR
μ^*	a	9	9	9	9	9	9	9	9	9	9	9	9	9	9	9
	m_{sDOY}	8	3	8	7	7	8	8	8	7	7	8	2	8	3	8
	b_{sDOY}	7	8	7	4	4	7	6	7	5	8	7	6	6	8	6
	m_{eDOY}	6	1	6	8	8	6	7	4	4	6	6	4	7	6	7
	b_{eDOY}	5	7	4	6	6	5	5	2	2	5	5	5	5	7	5
	m_{prod}	4	5	3	5	5	3	4	5	8	4	4	3	4	1	4
	b_{prod}	3	2	2	3	3	4	2	6	3	3	3	1	3	2	3
	P_{norm}	2	4	1	2	1	1	1	1	1	2	1	7	2	4	1
	P_{annual}	1	6	5	1	2	2	3	3	6	1	2	8	1	5	2
σ	a	9	9	8	9	9	9	9	9	9	9	9	9	9	9	9
	m_{sDOY}	7	3	9	6	5	8	8	2	6	7	8	1	7	3	8
	b_{sDOY}	8	7	3	3	7	6	6	5	8	7	6	6	6	8	6
	m_{eDOY}	6	1	6	8	8	4	5	7	4	6	6	4	8	7	7
	b_{eDOY}	5	7	4	7	7	6	7	1	1	5	5	5	5	6	5
	m_{prod}	4	4	3	5	6	3	4	5	8	4	4	3	4	1	4
	b_{prod}	3	2	2	4	4	5	3	8	3	3	3	2	3	2	3
	P_{norm}	2	5	1	2	1	1	1	4	2	1	1	7	2	5	1
	P_{annual}	1	6	5	1	2	2	2	3	7	2	2	8	1	4	2

Table S2 Model sensitivity analysis parameter ranks from Morris indices

The simulated Morris indices (mean m^* and standard division s) for the 9 input parameters for the 15 model taxa. The highest 4 ranking parameters are highlighted with warm colors, indicating a larger importance of the pollen emission simulation.

Comment:

More detail is needed on how the authors downscaled the climate data to a common grid. The link to the table in the supplementary material does not include spatial resolution for a number of models.

Response:

The link provided in the SI does not include the spatial resolution of all CMIP6 models, but it does include the resolution of the models we selected. The original model resolutions of the CMIP6 simulations used in this study are included in revised Table S3.

We also added text to explain the downscaling method in the Table S3 caption and in the methods section on lines 497-500.

Lines 497-500:

“Climate data are regridded to a 25km Lambert Conformal Conic projection⁴¹ using the Earth System Modeling Framework (ESMF) higher-order patch regridding method over the United States to match the spatial resolution of PECM..”

Comment:

I am guessing that the authors calculate pollen emissions for each species for each model and then calculate an evenly weighted multi-model average? The alternative approach is to calculate a multi-model average of the daily climate projections and then pass them through the pollen model. Both approaches are used commonly, however, the reader needs to know which approach was used here.

Response:

Thank you for pointing this out. We applied the first approach for analysis: “calculate pollen emissions for each species each model and then calculate an evenly weighted multi-model average.” We specific that in the revised manuscript:

Line 504-506:

“Pollen emissions are simulated using the meteorology data input from each CMIP6 model and then an evenly weighted multi-model average from 15 PECM simulations is calculated for analysis.”

Comment:

Results in Figure 4 would be more robust if they showed a multi-model average climate projection rather than results from a single model.

Response:

Thank you for noting this point. In the revised manuscript, we use results from multi-model average projection and update the figure and caption to reflect this change:

Fig.3: Simulated regional pollen seasonal magnitude and timing in the historical and future.

a-f, 20-year average time series of daily pollen emission flux ($grains\ m^{-2}d^{-1}$) of the six dominant individual tree taxa in DBL and the total DBL emission with (solid black line) and without (dashed black line) precipitation effects. (**a-c**) Historical (1995-2014) emission and (**d-f**) end of the century (2081-2100) emissions from multi-model average simulation for SSP 585. **g**, There are 5 geographic regions used in this study: Northeast (NE; 38-48° N and 70-100° W), Southeast (SE; 25-38° N and 70-100° W), Mountain (MT; 25-48° N and 100-116° W), California (CA; 25-40° N and 116-125° W) and Pacific Northwest (PNW; 40-48° N and 116-125° W). Three regions are selected for DBL daily pollen emission analysis: Northeast, NE (**a, d**); Mountain, MT (**b, e**); Pacific Northwest, PNW (**c, f**).

Reviewers' Comments:

Reviewer #1:

Remarks to the Author:

The authors have done a very thorough job addressing my comments. I thank them for this effort and have no further comments about the study. It's a very nice study and will make an important contribution to the literature.

Reviewer #2:

Remarks to the Author:

The authors have addressed my concerns to my satisfaction. They have improved on the manuscript by modifying the model to include a time-varying annual pollen production improves this analysis.

Reviewer #3:

Remarks to the Author:

I thank the authors for actioning all of my suggestions. While this has improved the paper, I do have some additional minor suggestions.

1. When describing the approach used to account for shifting plant distributions, ensure that it is clear to the reader that the GCAM data is forced by climate as well as landuse and that the climate data underpinning the GCAM projections is the same as what is being used in the scenarios with fixed plant distributions. In other words, you are comparing like with like. This is very unclear in the current text. It is important that there is indeed a match between climate models forcing the PECM and those used to force the distribution shifts in plant functional types.

2. While I would not have done the sensitivity analysis like the authors have done it, its addition is important. However, I do recommend that the authors describe the sensitivity analysis earlier in the paper, focusing on only its generality in the Discussion .

3. The Discussion is a bit light on. It would be good if the authors could give a bit more thought to the generality of their findings as well as the caveats of their approach. Regarding caveats the authors should discuss other ways for modelling future range shifts for their candidate species. I would be very surprised if the data is not available to build species-level ecological models and project them under climate and landuse scenarios i.e., using species distribution models.

Manuscript NCOMMS-21-14254-T

Response to reviewers

Dear reviewers,

We'd like to thank you again for reviewing the manuscript "Projected climate-driven changes in pollen emission season length and magnitude over the continental United States" for publication in *Nature Communications*. We appreciate the suggestions in the first round of review and are pleased to know that we addressed your concerns.

Our point-by-point response to the reviewers' comments and concerns is provided below, along with a tracked changed version of the manuscript that highlights all changes. We also produce a final version of the revised manuscript, with all line numbers included below referring to the final untracked version.

Response to reviewer 1:

Comment:

The authors have done a very thorough job addressing my comments. I thank them for this effort and have no further comments about the study. It's a very nice study and will make an important contribution to the literature.

Response:

Thank you for the positive comments! We appreciate the time and effort you gave to improving this manuscript.

Response to reviewer 2:

Comment:

The authors have addressed my concerns to my satisfaction. They have improved on the manuscript by modifying the model to include a time-varying annual pollen production improves this analysis.

Response:

Thank you so much! We appreciate your suggestions to develop climate-dependent production factors, as this greatly improved the manuscript.

Response to reviewer 3:

Comment:

I thank the authors for actioning all of my suggestions. While this has improved the paper, I do have some additional minor suggestions.

Response:

Thank you for the suggestions, they are addressed below.

Comment:

When describing the approach used to account for shifting plant distributions, ensure that it is clear to the reader that the GCAM data is forced by climate as well as landuse and that the climate data underpinning the GCAM projections is the same as what is being used in the scenarios with fixed plant distributions. In other words, you are comparing like with like. This is very unclear in the current text. It is important that there is indeed a match between climate models forcing the PECM and those used to force the distribution shifts in plant functional types.

Response:

We agree this would be a helpful clarification. The description of GCAM model is revised in the Method:

Line 501-503:

Methods:

“For the sensitivity test, we simulated the future maximum daily pollen emission both using historical (2015) and future (2100) PFT land cover data from GCAM-Demeter land use dataset⁴⁹, which is driven by the same climate forcing data used for PECM.”

Comment:

While I would not have done the sensitivity analysis like the authors have done it, its addition is important. However, I do recommend that the authors describe the sensitivity analysis earlier in the paper, focusing on only its generality in the Discussion.

Response:

Thank you for the comment, we edited the paper and described the sensitivity earlier in the manuscript. Please see the revised manuscript for the changes:

Line 177-184:

Main:

“As both pollen production and phenology have the potential to impact pollen emission maxima ($E_{\text{pol,max}}$), a sensitivity analysis (see Methods) indicates that the temperature-dependent regression and normalization parameters of pollen production (m_{prod} , b_{prod} , and P_{norm} in equation (5)) have the greatest impacts on the simulated pollen amount (Supplementary Table 2). For taxa with pollen season duration sensitive to temperature change (*Alnus*, *Platanus*, *Populus*, late-flowering *Ulmus*, C4 grass), the regression parameters of phenology can also become important. Overall, the magnitude of annual pollen production is one of the most important parameters in model simulation (P_{annual} in equation (5); Supplementary Table 2).”

We then describe the generality of our sensitivity analysis in the Discussion:

Line 347-352:

Discussion:

“The sensitivity analysis of model parameters (see Methods) indicates the dominant uncertainty is related to the pollen production and the climate-relevant production parameters, which is

derived from limited suite of field-based studies (Supplementary Table 1). More measurements across space and time could improve our understanding of pollen production and better constrain the model simulations.”

Comment:

The Discussion is a bit light on. It would be good if the authors could give a bit more thought to the generality of their findings as well as the caveats of their approach. Regarding caveats the authors should discuss other ways for modelling future range shifts for their candidate species. I would be very surprised if the data is not available to build species-level ecological models and project them under climate and landuse scenarios i.e., using species distribution models.

Response:

There is an incomplete sentence in the first comment (e.g., “The Discussion is a bit light on. It would be good if...”) and therefore we are not sure if all of the reviewer’s concerns were completely communicated. Therefore, we have addressed the comments that we are able to understand in full.

As requested by the reviewer, we added more discussion about the current available methods of species distribution shift modeling in the discussion:

Line 357-365:

Discussion:

“Finally, large uncertainties in future plant community shifts^{51,52} also limit the simulations of land cover change effects on pollen emission. Recent advances in species distribution modeling include the development of new approaches (e.g., regression-based and machine learning)⁵³, yet there are large uncertainties connected to climate change projection and biotic stresses (e.g., insects, fungi, bacterial)⁵³. Gridded taxa-specific land cover change data for multiple taxa over the entire CONUS is still lacking⁵⁴⁻⁵⁶. Our simulations using PFT-based land cover change data provide overall estimates of vegetation shifts, but the development of the spatially resolved taxa-specific land cover data over a large scale will be crucial to evaluate the effects of plant community composition change on future pollen emission.”

We have also added more text to the final paragraph of the Discussion section to highlight the generality of this work:

Line 376-382:

Discussion:

“Although land cover change has relatively smaller effects in our simulations suggesting that the climate drivers may be more important and occurring faster than the shifts in vegetation distribution, our approach does not account for the spatial shifts of individual taxa ranges, which will certainly be important to assess future regional pollen emission composition. This study provides an important predictive tool to start to investigate the consequences of climate change on the future plant communities and their corresponding health effects.”